# Shear localisation controls the dynamics of earthquakes

Fabian Barras [1] ✉ & Nicolas Brantut [2,3] ✉

Earthquakes are produced by the propagation of rapid slip along tectonic faults. The propagation dynamics is governed by a balance between elastic stored energy in the surrounding rock, and dissipated energy at the propagating tip of the slipping patch. Energy dissipation is dictated by the mechanical behaviour of the fault, which is itself the result of feedbacks between thermo-hydro-mechanical processes acting at the mm to sub-mm scale. Here, we numerically simulate shear ruptures using a dual scale approach, allowing us to couple a sub-mm description of inner fault processes and km-scale elastodynamics, and show that the sudden localisation of shear strain within a shear zone leads to the emergence of classical cracks driven by a constant fracture energy. The fracture energy associated to strain localisation is substantially smaller than that predicted in theoretical and numerical models assuming uniform shearing within the shear zone. We show the existence of a unique scaling law between the localised shearing width and the rupture speed. Our results indicate that earthquakes are likely to be systematically associated to extreme strain localisation.

Earthquake sources correspond to slip events dynamically propagating along faults. At crustal scale, faults can be viewed as two-dimensional surfaces, across which the displacement field is discontinuous. However, geological and experimental observations show that 'slip' across faults is the result of shear deformation across narrow layers of highly comminuted, transformed or partially melted rocks. In the shallow continental crust, fault core materials are often made of fine-grained siliclastic and clay gouges, with a porosity filled with pressurised water[1,2]. The dynamics of ruptures in crustal faults is controlled by the rheology of these water-saturated fault gouges.

During earthquakes, faults slide at elevated slip rates of the order of metres per second, which leads to dramatic weakening of fault gouge materials (e.g. chap. 2 of Scholz[3]). In dry materials, weakening is most likely controlled by the local temperature rise arising from dissipation of frictional work, combined with thermally activated rheology of the rock-forming minerals[2,4–9]. In the presence of fluids, an additional weakening mechanism is expected, due to the differential thermal expansion of the pore fluids and the solid pore space: upon heating, the fluid pressure rises, effective normal stress decreases and

the frictional strength drops. This so-called 'thermal pressurisation' mechanism, initially proposed by Sibson[10] as a temperature-limiting process in wet rocks, has been shown to produce realistic predictions for thermal evolution and energy dissipation during earthquakes[2,11], and is a potential candidate to explain some of the complexity observed in natural earthquakes[12] and the operation of plate boundary faults at low ambient stress[13,14].

The thickness of the actively deforming zone determines the shear heating rate and how easily fluids and heat diffuse away from the fault plane, and thus has a tremendous influence on the resulting rupture dynamics[11,13–16]. While geological and experimental observations can be used to constrain the thickness of actively deforming fault gouge material, the range of acceptable values spans more than 3 orders of magnitude, from fractions of millimetres to centimetres[2], and it is one of the key unknown that limits our ability to determine the efficiency of thermal weakening mechanisms in nature.

The influence of shear zone width on earthquake propagation is further complicated by the fact that this parameter is likely evolving during seismic slip: strain localisation is expected to be part of the

[1]The Njord Centre, Department of Physics, Department of Geosciences, University of Oslo, Oslo, Norway. [2]Department of Earth Sciences, University College London, London, UK. [3]GFZ Helmholtz Centre for Geosciences, Potsdam, Germany. ✉e-mail: fabian.barras@mn.uio.no; nicolas.brantut@gfz-potsdam.de

fault weakening process. Several mechanisms might be responsible for strain localisation during earthquake slip, including granular rearrangements and grain size reduction[17,18], shear heating coupled to thermal weakening[19], thermal pressurisation[20–22] and thermal decomposition[23,24]. In all cases, the strain localisation process is associated to a rapid reduction in shear strength, and we therefore expect strain localisation to exert a strong control on the overall dynamics of rupture.

Here, we demonstrate and quantify how strain localisation impacts rupture dynamics: we run dynamic rupture simulations, find a relationship between the rupture speed and the degree of strain localisation within the fault gouge and compute the portion of dissipated energy that controls the rupture propagation; a quantity that is often referred to as the *fracture energy* in analogy to the dynamics of fractures in brittle materials. We use the case of thermal pressurisation as a representative thermal weakening process that is compatible with seismological observations[2,11,14], and is known to spontaneously lead to strain localisation[21,22]. We argue that the interplay between rupture dynamics and strain localisation analysed here applies to most thermal weakening processes in rocks and engineering materials.

## Results

### Shear localisation and faulting
In this paper, we focus on spontaneous strain localisation that occurs over short time scales –coincident to the few seconds that dynamic rupture propagation lasts– and leads to the formation of sub-millimetric principal slip surfaces within fault cores as sketched in Fig. 1. Rice et al.[21] demonstrated that thermal pressurisation can produce instabilities in the rheology of fault gouge that lead to spontaneous strain localisation. Interestingly, a similar mechanical instability is associated to the localisation of deformation in various types of material rheology, including the formation of plastic shear bands in metallic alloys[25] or dense amorphous materials[26], as well as thermal runaway failure in visco-elastic materials[19]. Consequently, spontaneous strain localisation is expected to arise under a large variety of geological conditions but emerging from a generic feedback mechanism in the rheology of fault zones that we identify and summarise hereafter. As a general starting point for our analysis, let us consider slip on geological fault as the deformation distributed across a narrow pre-existing shear zone whose rheology relates the shear stress $\tau$ to a set of

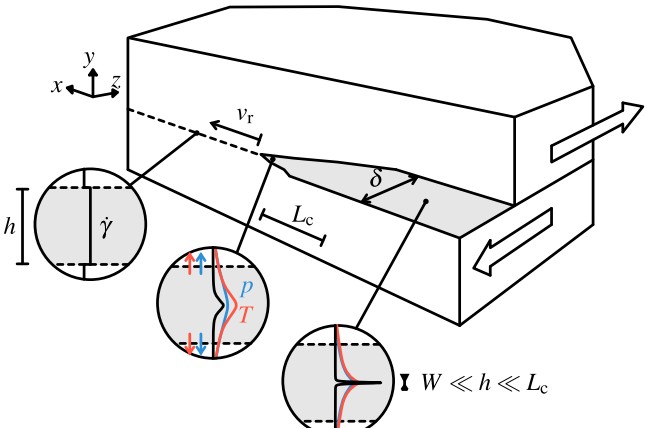

**Fig. 1 | Schematic of the dual scale setup governing the propagation of localised shear bands.** The dynamic rupture extends over large (kilometric) scales along the fault ($x-z$ plane), whereas the frictional strength is determined by solving a coupled diffusion problem, such as thermal pressurisation in this paper, where strain rate spontaneously evolves over submillimetre scales across the fault gouge (in the $y$ direction).

variables that includes the shear strain rate $\dot{\gamma}$:

$$\tau = f(\dot{\gamma}, \vartheta). \tag{1}$$

In the relationship above, $\vartheta$ is a diffusive quantity to which the rate of work produced by frictional shearing acts as a source term:

$$\dot{\vartheta} = \beta\tau\dot{\gamma} + \alpha\nabla^2\vartheta, \tag{2}$$

where $\nabla^2$ denotes the Laplace operator, $\alpha$ is a diffusivity and $\beta$ is analogous to the Taylor-Quinney coefficient[27] for the cases when $\vartheta$ corresponds to temperature. From the constitutive relationship (1), one can define

$$g'(\dot{\gamma}, \vartheta) = \frac{\partial f}{\partial \dot{\gamma}}, \tag{3}$$

which describes the rate-dependent rheology of the material. Natural examples include viscous creep of rocks at elevated temperature, granular material rheology[28] and rate-and-state friction. Similarly, one can define

$$h'(\dot{\gamma}, \vartheta) = \frac{\partial f}{\partial \vartheta}, \tag{4}$$

to describe the effect of $\vartheta$ on the material rheology. In practice, this diffusive quantity $\vartheta$ often corresponds to temperature and $h'$ describes thermal weakening effects. $\vartheta$ could also correspond to fluid pressure in a porous material whose strength is reduced by an increase in pore fluid pressure (following the concept of effective stress discussed later in Equation (7)). It can also account for the combined effect of pressure and temperature as in the case of thermal pressurisation that will be discussed later in this manuscript. If conditions (1) and (2) are met, a linear stability analysis (detailed in "Methods") demonstrates that uniform shearing at a given time $t = t_0$ becomes unstable if:

$$\frac{h'_0}{g'_0} < 0, \tag{5}$$

with $\{f_0, g'_0, h'_0\} = \{f, g', h'\}|_{t=t_0}$. Moreover, the analysis also shows that only perturbation wavelengths $\lambda$ greater than a critical wavelength are unstable:

$$\lambda > 2\pi\sqrt{-\frac{\alpha}{\beta f_0}\frac{g'_0}{h'_0}} \equiv \lambda_c, \tag{6}$$

which explains that such instability drives the localisation of shear strain down to some thickness $W_{loc} \sim \lambda_c/2$. Remarkably, this type of localisation instability can also arise within rate-strengthening materials ($g'_0 > 0$) providing that $h'_0 < 0$, as it is often the case with thermal weakening mechanisms. As a result, shear deformation concentrates over a thickness much smaller than the width of the shear zone, which leads to a substantive drop of the associated shear stress. In this work, we quantitatively investigate and discuss how the drop of shear stress caused by spontaneous strain localisation across the fault zone drives the propagation of failure and the rapid acceleration of slip along the fault plane at the origin of earthquakes.

### A multi-scale model of faulting
Next, we focus on the process of thermal pressurisation, which has been shown to be a realistic dynamic weakening mechanism[2,11,14] and that undergoes the localisation instability outlined above[21,22]. In this case, the diffusive variable $\vartheta$ corresponds to pore fluid pressure $p$ that affects the effective normal stress ($\sigma_n - p$) in the shear zone and, thereby, its shear strength together with a rate-dependent friction

coefficient:

$$\tau(\dot{\gamma}, p) = f_{\text{rsf}}(\dot{\gamma})(\sigma_{\text{n}} - p). \tag{7}$$

In Equation (7), we adopt the rate-strengthening rheology $f_{\text{rsf}}(\dot{\gamma})$ of Platt et al. [22] detailed in "Methods". Moreover, fluid transport across the shear zone is governed by a coupled system of thermal and hydraulic diffusion equations (see Equations (33) in 'Methods'). This thermo-hydraulic coupling is caused by the different compressibilities and thermal expansivities of the solid matrix and the pore fluid, and describes the change in pore pressure produced by local temperature variations in the gouge. Platt et al. [22] solved numerically the dynamic evolution of strain rate along a one-dimensional shear zone under an imposed and constant relative slip rate $V$ measured between the top and the bottom of the gouge layer. They showed that this one-dimensional setup produces the localisation instability and that the resulting localisation of shear strain rate stabilises to a finite width that can be well approximated by

$$W_{\text{rsf}}(V) \simeq 6.9 \frac{A\rho c}{\Lambda f_{\text{c}}} \frac{\left(\sqrt{\alpha_{\text{hy}}} + \sqrt{\alpha_{\text{th}}}\right)^2}{V(f_{\text{c}} + 2A)}, \tag{8}$$

where $\alpha_{\text{hy,th}}$ correspond respectively to the hydraulic and thermal diffusivities, $\rho c$ is the heat capacity of the gouge material, $\Lambda$ is the thermal pressurisation parameter that describes the change of pore fluid pressure caused by an increase in temperature in the gouge. The characteristic shear strength

$$\tau_{\text{c}} = f_{\text{c}}(\sigma_{\text{n}} - p_0) = f_{\text{rsf}}(V/h)(\sigma_{\text{n}} - p_0) \tag{9}$$

is a function of the initial uniform strain rate $\dot{\gamma} = V/h$ and background pore pressure $p_0$. In Equation (8), the constant $A$ corresponds to the rate-strengthening coefficient that describes the 'direct' response of the shear zone to a change in strain rate similar to standard rate-and-state models (see Equation (34) in 'Methods'). Platt et al. [22] reported that the shear stress within the gouge initially decays exponentially following the adiabatic undrained prediction of Lachenbruch[29]:

$$\tau_{\text{adiab}}(\delta; V) = \tau_{\text{c}} \exp\left(-\frac{f_{\text{c}}\Lambda}{h\rho c}\delta\right) \equiv \tau_{\text{c}} \exp(-\delta/\delta_c). \tag{10}$$

During sliding, heat and pressure are produced across the thickness of the layer of sheared gouge. The adiabatic undrained regime lasts until diffusive heat and fluid transport into the surrounding wall rock becomes effective. At the same time, the development of gradients in pressure and temperature across the layer of gouge can trigger strain localisation following the mechanism described in the previous section. In the one-dimensional configuration studied by Platt et al. [22], the adiabatic-undrained prediction (10) provides a good approximation of the initial stage preceding the localisation instability, which typically arises for cumulative amount of slip in the range of the characteristic slip distance $\delta_c$. After strain localisation, the width of the actively sheared region $W_{\text{loc}}$ becomes much smaller than the diffusion length scale such that the shear stress evolution is well approximated by the *slip-on-a-plane* solution based on the assumption that $\dot{\gamma}(y) = V\bar{\delta}(y)$, with $\bar{\delta}$ being the Dirac delta distribution. In this situation, the shear stress across the shear zone is only function of the imposed slip rate $V$ and the accumulated slip $\delta = Vt$[2,30]:

$$\tau_{\text{sp}}(\delta; V) = \tau_{\text{c}} \exp\left(\frac{\delta}{L^*}\right) \text{erfc}\left(\sqrt{\frac{\delta}{L^*}}\right), \tag{11}$$

where

$$L^*(V) = \frac{4}{f_{\text{c}}^2}\left(\frac{\rho c}{\Lambda}\right)^2 \frac{\left(\sqrt{\alpha_{\text{hy}}} + \sqrt{\alpha_{\text{th}}}\right)^2}{V}. \tag{12}$$

The traction-versus-slip evolutions (10) and (11) provides two end-member predictions of the rheology of a one-dimensional layer of gouge from adiabatic simple shear flow to highly localised deformation. Both are characterised by a monotonous exponential decay of the shear stress with slip, which results in a monotonic increase in the frictional dissipation with slip. The integration of the *breakdown work* $E_{\text{BD}}$ is of seismological interest and corresponds to the energy dissipation measured on top of residual friction as the stress in the shear zone weakens toward residual strength:

$$E_{\text{BD}}(\delta) = \int_0^\delta \left(\tau(\delta') - \tau(\delta)\right) \text{d}\delta'. \tag{13}$$

Whereas the evolution of $E_{\text{BD}}(\delta)$ integrated from these one-dimensional predictions successfully captures the scaling of the breakdown work inverted from seismological observations[2,11], its contribution to the propagation of the earthquake rupture remains unclear. Firstly, predictions of rupture dynamics often rely on the dynamic fracture theory and on the possibility to isolate a very small region around the rupture tip (*the process zone*) and associated near-tip energy dissipated in breaking the fault zone strength (*the fracture energy*). Secondly, the slip rate across the gouge evolves rapidly near the propagating rupture tip, and is dynamically coupled to the drop of shear stress, which is not captured by the assumptions used to construct the one-dimensional predictions (10) and (11).

In this context, we aim to analyse the coupling that exists between strain localisation, slip acceleration and rupture dynamics in a simple faulting geometry that is sufficient to capture its key physical aspects. Exploring the interplay between strain localisation and rupture dynamics is a challenging dual-scale problem: it requires solving for heat and fluid diffusion at the scale of the fault core (from millimetres to centimetres in natural fault zones) together with the elastodynamics governing the propagation of the earthquake rupture along the fault (elastic waves moving at kilometres per second in crustal rocks). We follow Noda et al. [13] and take advantage of the separation of scale to solve thermal pressurisation only across the fault (along the $y$ axis in Fig. 1). We consider a planar fault within an infinite linear elastic medium sliding in anti-plane shear (mode III). Our numerical model brings two important differences from the standard earthquake simulations with thermal pressurisation (e.g. SCEC benchmark TPV105, Harris et al. [31]). Firstly, the profile of strain rate across the shear zone is not imposed but evolves dynamically through the rupture event. Secondly, the constitutive law (34) has no additional weakening mechanism (e.g. flash heating or rate-and-state), such that it directly relates the shear stress to the local strain rate within the gouge—and not to the macroscopic slip rate $V$. In this configuration (Fig. 1), the long-range elastodynamics couples the shear traction along the fault $\tau(x, t)$ to the strain rate in the shear zone and can be expressed by a boundary integral formulation[32]:

$$\tau(x, t) = \tau_{\text{b}} - \frac{\mu}{2c_{\text{s}}} \int_{-h/2}^{h/2} \dot{\gamma}(x, y, t) \, \text{d}y + \phi(x, t). \tag{14}$$

In the equation above, $\mu$ is the shear modulus, $c_{\text{s}}$ the shear wave speed of the linear elastic medium surrounding the shear zone and $\tau_{\text{b}}$ represents the far-field background stress. The integral on the right-hand side describes the instantaneous local stress change due to variations of the strain rate profile within the shear zone, whereas $\phi$

accounts for the dynamic interactions between different regions along the fault. Further details about Equation (14) and its numerical evaluation are provided in "Methods". Equation (14) leads to the definition of a characteristic seismic slip rate $V_c$ and associated characteristic strain rate $\dot{\gamma}_c$ as

$$\dot{\gamma}_c = \frac{V_c}{h} = \frac{2c_s\tau_b}{h\mu}, \tag{15}$$

which are used in the remainder of the paper together with the related characteristic shear strength $\tau_c$ (9). The elastodynamic equation (14) together with the rheology equation (7) couple the strain rate in the shear zone $\dot{\gamma}$ to the shear stress $\tau$ and allow for the implementation of a dual-scale coupled numerical scheme that solves the rupture elastodynamics along the shear zone together with pressure, temperature and strain rate evolutions within the shear zone. The details of our coupled two-dimensional numerical scheme are given in "Methods". The proposed multi-scale model rests upon the separation of scales shown in Fig. 1 that is relevant in the context of this paper focusing on the propagation dynamics of seismic rupture. The separation of scales is justified because the length scale of the diffusive processes over the duration of the seismic rupture is much smaller than the characteristic length scale over which stress and slip rate change along the fault plane, typically of the order of a few metres. Hydraulic diffusivities in fault gouge are typically of the order of few millimetres square per second[2], which leads to diffusion lengths of the order of millimetre to centimetre for rupture event tens of seconds in duration. During the nucleation stage, this assumption might not always hold—such as in the case of marginally pressurised fault zone discussed in Garagash and Germanovich[33], Bhattacharya and Viesca[34].

**Rupture dynamics driven by shear localisation**

In our simulations, the shear zone is initially creeping at aseismic slip velocity and, at time $t = 0$, failure is nucleated by rising the pore pressure near the centre of the fault $x = 0$ over a region larger than the nucleation length but much smaller than the whole length of the domain (further details of nucleation procedure and parameter values are given in "Methods"). Initially, acceleration of slip is mostly concentrated in the nucleation region, followed by a rapid lateral rupture propagation whereby the slip rate increases in an expanding region beyond the initial nucleation patch, concomitantly with a shear stress drop linked with thermal pressurisation of pore fluid inside the gouge and intense strain localisation (Fig. 2). Rupture acceleration coincides with larger slip velocities and stress drop at the tip (Fig. 3a–c) and more intense localisation of shear deformation across the gouge where up to four orders of magnitude larger strain rate concentrates on less than five percent of the thickness of the shear zone (Fig. 3b). Interestingly, the peak slip rate and drop of shear stress measured at different positions along the fault arise for the same amount of slip and coincide with intense strain localisation (Fig. 3b, c). The amount of slip required to trigger localisation is similar to the one reported from one-dimensional simulations under imposed velocity and is in the order of magnitude of $\delta_c$ (see Figs 9 and 10 of Platt et al. [22]). Remarkably, this observation enables us to apply the one-dimensional theory discussed in the previous section to build predictions of the shear zone dynamics after strain localisation. For instance, the slip-on-a-plane solution described in Equation (11) can be used to capture the magnitude of the shear stress reached immediately after strain localisation $\tau \approx \tau_{sp}(\delta = \delta_c; V = V_{tip})$, with $V_{tip}$ being the slip rate observed at the rupture tip (see Fig. 3c and related caption). Moreover, once the localisation instability arises, the thickness of actively strained material at various positions along the interface closely follows a single $W_{rsf}(V)$ curve, which follows the prediction given in Equation (8). The dynamics reported in Fig. 3c demonstrate how frictional weakening during the rupture is caused by two successive mechanisms operating

over different magnitudes of slip: (1) The rapid localisation of strain that creates an abrupt drop of stress in the shear zone for slip $\delta \approx 0.3\delta_c$ and (2) co-seismic thermal pressurisation that progressively prolongs frictional weakening over larger values of slip. During this second stage, a progressive delocalisation of strain rate is observed within the shear zone (Fig. 3). Next, we quantitatively demonstrate that the propagation of the rupture is mainly governed by the first mechanism, i.e. by dynamic strain localisation.

Let us analyse snapshots of the propagating rupture and the near-tip evolution of the macroscopic and microscopic mechanical variables (Fig. 4). Ahead of the propagating tip (point A), the shear zone is creeping with uniform shear strain rate. As the rupture approaches, the strain rate builds up uniformly across the gouge (point B) until the localisation instability arises (point C) together with a rapid increase in macroscopic slip rate $V$ and abrupt drop of shear stress $\tau$. In the wake of the rupture (point D), the profile of strain rate across the gouge progressively delocalises, following the decay of the macroscopic slip rate given by the prediction $W_{rsf}(V)$ shown in Fig. 3b. The near-tip evolution of $V$ and $\tau$ is reminiscent of the singular solutions at the tip of a dynamic fracture[35]. Defining $\alpha_s^2 = 1 - v_r^2/c_s^2$, the analogy to linear elastic fracture mechanics (LEFM) can be quantitatively tested by rescaling the near-tip stress and slip rate according to

$$\Delta\tau = \tau(x - x_{tip}) - \tau_{res} = \frac{\mu\alpha_s}{2v_r}V(x_{tip} - x) = \frac{K}{\sqrt{2\pi|x - x_{tip}|}} \tag{16}$$

and fitting the dynamic fracture solution (16) following the procedure of Barras et al. [36], respectively ahead of the rupture for $\tau(x - x_{tip})$ and in the wake of the rupture for $V(x_{tip} - x)$. The stress intensity factor $K$, residual stress $\tau_{res}$ and position of the tip $x_{tip}$ are the free parameters that are fitted simultaneously to match the near-tip increase of $\tau$ ahead of the rupture tip and decrease of $V$ behind the rupture tip.

The good agreement with dynamic fracture solution (dashed blue curves in Fig. 4) confirms the crack-like nature of the simulated rupture process near the tip of the slipping patch. The strain rate profiles (top panels of Fig. 4) shows that dynamic strain localisation arises between points B and C and the inverted position of the rupture tip confirms that this region precisely corresponds to the *process zone* of the propagating rupture. Moreover, such an agreement allows us to use the inverted value of $K$ and invoke the crack-tip energy balance to compute the *fracture energy*

$$G_c = \frac{K^2}{2\mu\alpha_s}, \tag{17}$$

which corresponds to the part of near-tip dissipated energy that governs the propagation of the rupture and that is missing in the one-dimensional description of thermal pressurisation. In seismology, extracting the fracture energy of natural earthquakes still eludes the resolution of seismic inversions, such that the *breakdown work* (see Equation (13)) is often used as a proxy for $G_c$ and integrates the excess of work on top of residual friction[37,38]. In our numerical simulations, the integration of $E_{BD}$ at different locations along the interface reveals a clear plateau over an order of magnitude in slip (Fig. 5). Following the theoretical work of Brener and Bouchbinder[39], such a crossover between two regimes in the breakdown work of frictional rupture indicates the portion of $E_{BD}$ that corresponds to $G_c$ in analogy to the small-scale yielding condition in the dynamics of fracture in brittle materials. This condition requires that the size of the process zone should be much smaller than other representative length scales of the elastic system for the propagation dynamics to be governed by the fracture energy $G_c$. In our geometry, the small-scale yielding condition is twofold: perpendicular to the fault plane, where it is imposed by the geometry of the problem and the small thickness of the layer $h \ll L_c$ (Fig. 1), and along the fault where the smallness of the

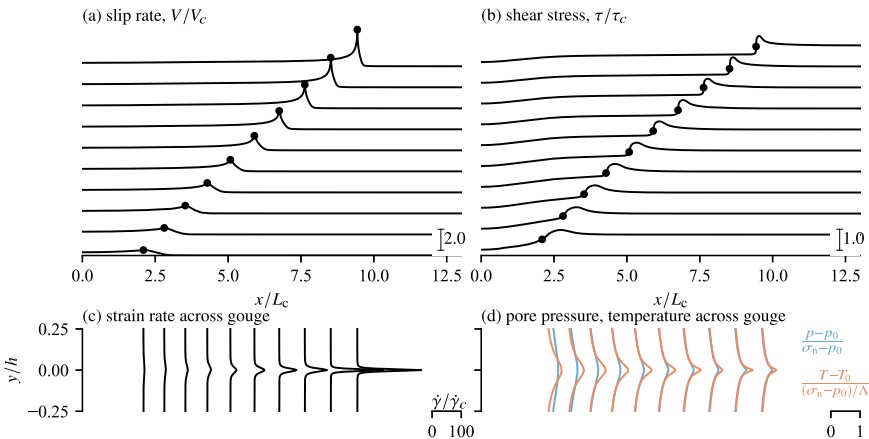

**Fig. 2 | Dynamic rupture driven by shear localisation simulated with the coupled model.** The top (**a**, **b**) respectively present snapshots at different times of the longitudinal profile of slip rate and shear stress during which the rupture accelerates from sixty to about ninety percent of the shear wave velocity. Note that the simulated domain is symmetric with respect to the nucleation position $x = 0$ such that another rupture tip moves toward the negative positions. The

bottom (**c**, **d**) present the profile of strain rate $\dot{\gamma}$, pressure $p$ and temperature $T$ at the positions along the interface with the most intense localisation highlighted by black dots in (**a**, **b**). See "Methods" and Table 1 for further details on the dimensional analysis behind this coupled problem and the dimensionless scaling used to plot the data in the different panels.

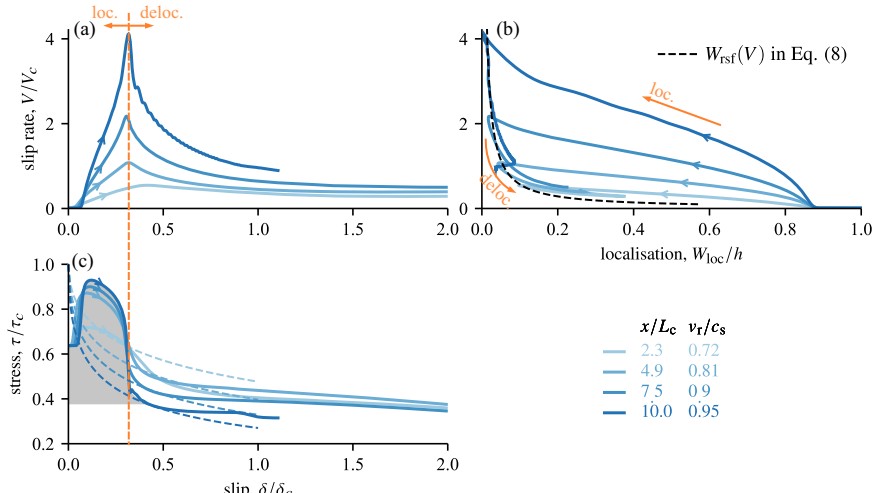

**Fig. 3 | Time evolution of the interfacial variables at different locations along the interface during the dynamic rupture shown in Fig. 2.** Line colours relate to the positions along the interface $x/L_c$ and the associated propagation speeds of the rupture $v_r/c_s$, whereas the arrows point to the direction of forward time evolution. The orange arrows demarcate sequences of strain rate localisation vs. delocalisation within the shear zone. Slip rate (**a**) and shear stress (**c**) versus slip revealing how the peak slip rate is associated to abrupt stress drop and arises at the same amount of cumulated slip $\delta_{loc} \approx 0.3\delta_c$. **b** Slip rate versus width of strain rate localisation $W_{loc}$ measured from the $\dot{\gamma}(y)$ profiles following the procedure shown in Fig. 3 of ref. 22.

The different post-peak delocalisation trajectories collapse along a single prediction given in Equation (8). The grey area in (**c**) sketches the dissipated fracture energy $G_c$ that balances the energy released by the rupture propagation, which is estimated from the bulk elastodynamics (see Fig. 4 and related explanations in the text). The dashed lines in panel (**c**) correspond to the prediction $\tau_{sp}(\delta; V_{tip})$ and gives a good prediction of the residual shear stress reached immediately after strain localisation. The slip rate at the rupture tip $V_{tip}$ is approximated by $V$ at the mid-time between the peaks in shear stress and in slip rate. (A more precise definition of the tip position is discussed and computed later in the context of Fig. 4).

process region depends on the weakening behaviour. If a clear plateau is observed in the breakdown work profile (Fig. 5), the first weakening stage relevant for $\delta \ll \delta_c$ arises over a *process region* near the rupture tip much smaller than that relevant for the second weakening stage, effective for $\delta \gg \delta_c$. Such a plateau in the profile of $E_{BD}(\delta)$ indicates that the rupture dynamics differs only slightly from the predictions of dynamic fracture theory and that the rupture propagation is well described by the fracture energy $G_c$[39], which explains the good agreement observed between the stress and velocity fields near the rupture tip and that predicted by dynamic fracture theory. Moreover, the first near-tip weakening stage and associated dissipation $G_c$ arise between points B and C in Fig. 4 and is then directly correlated to the localisation process.

We have then two independent measurements of the portion of dissipated energy that controls the rupture propagation—from the near-tip singularity and from the integration of the breakdown work. Remarkably, we find that these independent estimates of $G_c$ agree on the same constant value (grey horizontal line in Fig. 5), which is an additional proof of the crack-like nature of the rupture dynamics. Furthermore, the observed plateau in $E_{BD}$ is clearly associated to the rapid stress drop caused by localisation instability (see $\tau(\delta)$ profile in Fig. 3c) and brings further support to the fact that rapid strain localisation is the driving mechanism of the propagating rupture. Importantly, the magnitude of $G_c$ associated to strain localisation is more than five times smaller than that expected from uniform shearing under adiabatic undrained conditions ($\sim \tau_c \delta_c$). Such gap cannot be

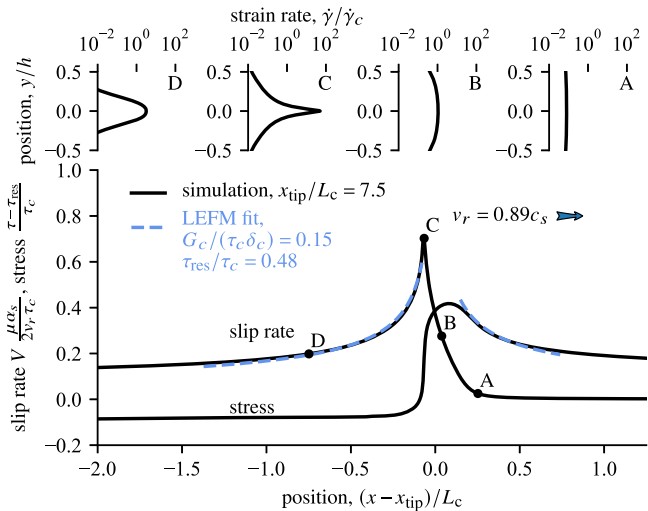

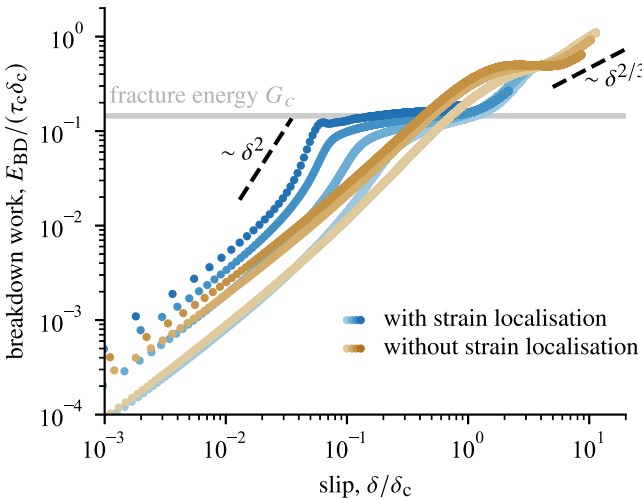

**Fig. 4 | Snapshot near the tip of the propagating rupture shown in Fig. 2.**
Bottom panel presents the spatial evolution of the shear stress and slip rate, which are simultaneously fitted by the fracture mechanics prediction shown by the dashed blue curve. (See the main text for details on the fitting procedure). Top panels show the strain rate profile across the shear zone observed at the instants A, B, C and D corresponding to the black dots in the bottom panel.

**Fig. 5 | Breakdown energy integrated from the $\tau$ versus $\delta$ profiles at different positions along the fault, identical to the positions used in Fig. 3 (blue dots) and for another simulation with the same parameters but neglecting the possibility for strain localisation (brown dots).** The rapid loss of stress caused by strain localisation creates an horizontal plateau whose associated magnitude is well predicted by the fracture energy inverted from the dynamic fracture fit shown in Fig. 4 and highlighted here by the horizontal grey line. The emergence of such near-tip dissipation is absent from the dynamics of rupture simulated with model neglecting strain localisation.

bridged by replacing $h$ in the adiabatic undrained prediction (10) by the smaller thickness of the actively slipping layer, notably because the latter is varying over several order of magnitude (Fig. 6) during the seismic rupture, whereas the value of $G_c$ stays rather constant. The inversion of the energy release rate at different positions along the interface (see Supplementary Fig. 1) or for simulations with different sets of parameters leads to similar observations and conclusions providing that dynamic strain localisation arises.

The interplay between strain localisation and rupture dynamics can be further established by relating the thinnest localisation width observed at a given location along the interface to the local speed of the rupture (Fig. 6). Following the behaviour reported in Fig. 3a, the dynamic stress drop caused by the localisation instability can be estimated by the slip-on-a-plane solution: $\Delta\tau \approx \tau_c - \tau_{sp}(\delta_c; V_{tip})$. The slip rate near the tip and the dynamic stress drop are also related by the elastodynamic relation (see Equation (16) and Fig. 4), which leads to the following implicit relationship between $v_r$ and $V_{tip}$:

$$V_{tip} = \frac{2v_r}{\mu\alpha_s}\left(\tau_c - \tau_{sp}\left(\delta_c; V_{tip}\right)\right), \tag{18}$$

which can be combined to the solution of Equation (8) to provide an approximate relationship between $v_r$ and the degree of localisation $W_{loc}$:

$$W_{loc} = W_{rsf}(V_{tip}). \tag{19}$$

In establishing relation (19), we assumed that the width $W_{rsf}$, originally derived assuming a constant slip rate with initially uniform pore pressure and temperature, also holds for the minimum localised width during slip with variable slip rate and complex pressure and temperature history. Despite this considerable simplification, we find that the estimate (19) is in reasonable agreement with the minimum localised width computed during rupture propagation in a range of numerical simulations with variable local rupture speeds (Fig. 6). The approximate validity of Equation (19) implies that strain localisation adapts rapidly compared to the slip rate evolution within the process zone, and is weakly sensitive to the initial conditions in terms of

pressure and temperature. Thus, the minimum shear zone width during rupture propagation is a good indicator of the peak slip rate, and thus of the potential dynamic rupture velocity.

The results analysed above (Figs. 3–6) correspond to simulations with rupture dynamics that are representative and similar to other rupture events that can be produced by our artificial dynamic nucleation procedure with different sets of parameters once the size of the nucleation patch is large enough. In our coupled system, the nucleation length is then intimately related to the propensity of shear strain to localise and its associated fracture energy.

## Dynamic strain localisation leads to the emergence of crack-like dynamics

Our simulations demonstrate that strain localisation produces a dramatic loss of stress-bearing capacity of shear zones that can create and sustain earthquake rupture, in addition to other weakening mechanisms that impact the long-term strength of the fault. The abrupt drop of shear stress produces an accelerating crack-like rupture in agreement with the predictions of dynamic fracture theory[35]. Notably, the rupture is driven by a well-defined fracture energy that corresponds to the near-tip dissipation during the localisation process. Such a behaviour is in contrast with that of ruptures driven by thermal pressurisation only, as in models neglecting internal strain localisation[11], for which breakdown work uniformly increases with increasing slip without the possibility to isolate $G_c$ from the remainder of breakdown dissipation. To demonstrate quantitatively the impact of strain localisation, we implement and run another type of simulations that uses the same standard parameter set but enforces strain over a constant width (without the possibility of spontaneous strain localisation). The implemented model is detailed in "Methods" and follows the common approach to simulate earthquakes driven by thermal-pressurisation[13]. The two models have the same asymptotic regimes at small and large slip $\delta$ following the prediction of Viesca and Garagash[11] (Fig. 5), but only the simulation that allows strain localisation features a clear separation of scales between the two regimes. As a direct consequence, it is no longer possible to isolate the near-tip fracture energy $G_c$ for the

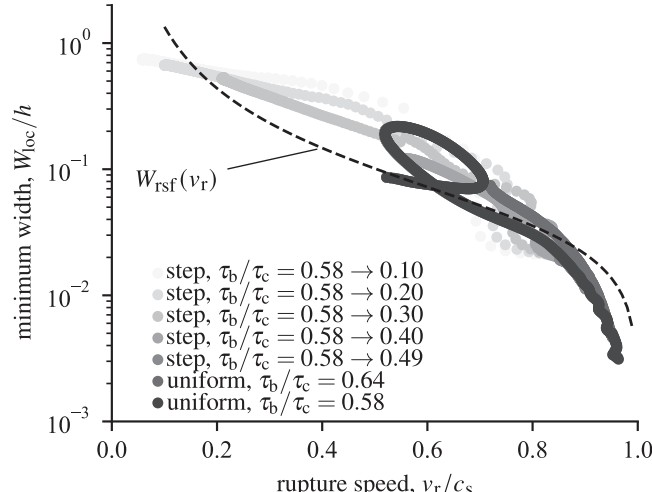

**Fig. 6 | Minimum strain rate localisation width $W_{loc}$ versus instantaneous rupture speed $v_r$ computed during rupture propagation for several simulations using the same parameters but different background stresses.** Heterogeneous simulations with steps in background stress were conducted with an initial plateau of $\tau_b/\tau_c = 0.58$ around the nucleation zone, and a sharp drop down to a smaller value at position $x/L_c = \pm17.75$ away from the centre of the nucleation region. In heterogeneous simulations, rupture speeds may vary nonmonotonically during propagation, initially increasing and subsequently decreasing when encountering a large downstep in stress. Regardless of the details of the dynamics, the relationship between peak localised width and rupture speed is well approximated by the theoretical prediction proposed in Equation (19).

simulation neglecting strain localisation. A salient implication is that the rupture dynamics significantly deviates from dynamic fracture predictions (see Supplementary Fig. 2). Consequently, dissipation over the entire crack length impacts the rupture propagation leading to larger fracture energy that increases with the rupture size (see Brener and Bouchbinder[39] for further discussion). In short, strain localisation causes both the embrittlement (lower values of fracture energy) of fault zones and the emergence of rupture crack-like dynamics in agreement with the prediction of LEFM. We expect therefore that dynamic strain localisation plays a major role in the propagation of earthquake ruptures.

## Discussion

Strain localisation within a preexisting gouge material is strongly correlated to the dynamics of fault slip, and specifically to the rupture speed (Fig. 6). The degree of strain localisation increases with increasing rupture speed, with a narrowing of the deformed, heated and pressurised region, approaching 1/1000th of the initial shear zone width. Despite the complexity of the problem, quantitative estimates can be obtained by a simple analytical approximation (Equation (19)) adapted from Platt et al. [22], so that the original predictions for peak localised width listed in Rice et al. [21], Platt et al. [22] still apply. Ideally, we could use the relationship between width and rupture speed depicted in Fig. 6 to interpret the localisation features observed in the geological record in terms of rupture dynamics. However, strain localisation in rocks is not exclusively associated with dynamic ruptures and fast slip rates[40], and only careful micro- and nano-structural studies can be relied upon to determine the seismic nature of geological structures, notably via detection of features characteristic of frictional heating[41]. Keeping this caveat in mind, our results highlight that the degree of strain localisation may be used as a complementary indicator of seismic slip: indeed, simulations leading to dynamic ruptures are always associated with strong localisation, with typical width in the sub-millimetre range[26].

Importantly, the multi-scale model implemented in this work allows us to demonstrate quantitatively how localisation exerts a pivotal control on the propagation of the earthquake rupture. In our model, thermal pressurisation is the constitutive law promoting the shear localisation instability, but a broad kind of material rheology have been shown to feature similar spontaneous localisation instability. We expect that our results can be generalised to other type of localisation instability arising within shear zones with no loss of generality. If the conditions (5) and (6) are fulfilled, a localisation instability can develop and lead to an abrupt drop of shear stress, which should lead to the emergence of a well-defined near-tip fracture energy and crack-like rupture. Importantly, shear localisation can produce and sustain rupture in shear zones having a rate-strengthening rheology ($g_0' > 0$) often interpreted as a token of stability and aseismic slip. Far from the tip, any diffusion-driven weakening leads to $E_{BD} \sim \delta^{2/3}$ at large slip[42]. Therefore, the behaviour summarised in Fig. 5 is expected to arise for any type of localisation-driven rupture, including those where the rheology is controlled by temperature, such as superplasticity[6,8,9,43]. Indeed, simulations of high speed deformation in metals, which are also rate-hardening and temperature-sensitive, tend to exhibit similar characteristics, with the emergence of a localisation-driven dissipation at the edge of propagating shear bands[44].

Our work demonstrates how localisation instabilities arising across a creeping shear zone create an abrupt drop of shear stress that promotes the propagation of classical dynamic ruptures over large distances along the shear zone. Whether frictional systems are governed by classical fracture mechanics or by nonlinear friction is an important and debated question in geophysics[14,38,45,46]. Strain localisation is an abrupt structural weakening mechanism that provides a clear separation between the process zone and the interior of the slipping patch, hence justifying the small-scale yielding hypothesis. However, the relative simplicity of the rupture tip behaviour does not preclude any complexity of the overall rupture style. Away from the rupture tip, thermal and hydraulic diffusion and strain delocalisation maintain a slow decay of the shear stress, which is prone to impact how earthquake ruptures stop[46]. The present results motivate further development of the proposed multi-scale approach to investigate the impact of strain localisation on other rupture modes, such as slow fronts[47] or pulse-like ruptures[48], that can arise under different boundary and initial conditions at nucleation.

## Methods
### Linear perturbation analysis
Let us consider a shear zone as in Fig. 1 initially creeping under imposed shear stress $\tau_0 = f_0$ and uniform strain rate $\dot\gamma_0(t)$ and field $\vartheta_0(t)$ conditions across the shear zone following the constitutive Equations (1) and (2):

$$\partial_t \vartheta_0 = \beta f_0 \dot\gamma_0, \tag{20}$$

with $\partial_t$ denoting a partial time derivative. At $t = t_0$, small perturbations to the uniform configuration are introduced such that the evolution of the three variables of interest can be written as

$$\begin{cases} \tau(y,t) = f_0 + \tau_1(y,t) \\ \dot\gamma(y,t) = \dot\gamma_0(t) + \dot\gamma_1(y,t) , \\ \vartheta(y,t) = \vartheta_0(t) + \vartheta_1(y,t) \end{cases} \tag{21}$$

with $\{\tau_1/f_0, \dot\gamma_1/\dot\gamma_0, \vartheta_1/\vartheta_0\} \ll 1$. Using the definitions of $g_0'$ and $h_0'$ given in the main text (3), (4), the constitutive law governing the shear zone rheology can be expanded as

$$f_0 + \tau_1 = f_0 + g_0' \dot\gamma_1 + h_0' \vartheta_1 + \mathcal{O}(\dot\gamma_1^2, \vartheta_1^2), \tag{22}$$

and further simplified as

$$\tau_1 = g_0' \dot{\gamma}_1 + h_0' \vartheta_1, \tag{23}$$

neglecting the higher-order terms. The diffusion equation can be similarly linearised

$$\partial_t \vartheta_0 + \partial_t \vartheta_1 = \beta(f_0 \dot{\gamma}_0 + f_0 \dot{\gamma}_1 + \tau_1 \dot{\gamma}_0) + \alpha \partial_y^2 \vartheta_1 \tag{24}$$

and simplified further using the uniform shear solution (20). The remaining terms write

$$\partial_t \hat{\vartheta}_1 = \beta(f_0 \dot{\gamma}_1 + \hat{\tau}_1 \dot{\gamma}_0) - 4\pi^2 k^2 \alpha \hat{\vartheta}_1, \tag{25}$$

using the following spectral decomposition

$$\{\hat{\tau}_1, \hat{\dot{\gamma}}_1, \hat{\vartheta}_1\}(k,t) = \int_{-\infty}^{\infty} \{\tau_1, \dot{\gamma}_1, \vartheta_1\}(y,t) e^{-2\pi i k y} \, \mathrm{d}y. \tag{26}$$

From the conservation of momentum and neglecting inertia through the thickness of the shear zone ($\partial_y \tau = 0$), one has $\hat{\tau}_1 = 0$ such that Equation (23) writes

$$\hat{\dot{\gamma}}_1 = -\frac{h_0'}{g_0'} \hat{\vartheta}_1. \tag{27}$$

Equation (25) becomes

$$\partial_t \hat{\vartheta}_1 = -\hat{\vartheta}_1 \left( \beta f_0 \frac{h_0'}{g_0'} + 4\pi^2 k^2 \alpha \right) \tag{28}$$

and has the following solution

$$\hat{\vartheta}_1 = \hat{\vartheta}_0 \exp \left( (t - t_0) \left( -4\pi^2 k^2 \alpha - \beta f_0 \frac{h_0'}{g_0'} \right) \right), \tag{29}$$

with $\hat{\vartheta}_0$ being the initial value of $\hat{\vartheta}_1$ at $t = t_0$. Consequently, uniform creeping across the shear zone is stable if

$$k^2 > -\frac{\beta f_0}{4\pi^2 \alpha} \frac{h_0'}{g_0'}, \tag{30}$$

which leads to the conditions (5) and (6) in the main text.

## Model

The longitudinal problem is solved assuming that the shear zone lays between two semi-infinite half-spaces. The elastodynamics is then solved using a boundary integral formulation that relates the traction $\sigma_{yi}(x, y = \pm h/2, t)$ acting within the shear zone to the respective displacements of each surfaces of the surrounding elastic wall rock $u_i(x, y = \pm h/2, t)$. We consider homogeneous elastic properties and antiplane shear conditions, such that the shear traction in the shear zone $\tau(x, t) = \sigma_{yz}$ can be related to the differential slip $\delta(x, t) = u_z(x, y = h/2, t) - u_z(x, y = -h/2, t)$ and slip rate $V(x, t) = \dot{u}_z(x, y = h/2, t) - \dot{u}_z(x, y = -h/2, t)$ following Equation (14), in which $\phi$ accounts for the non-local elastodynamic interactions and is evaluated in the Fourier domain as:

$$\Phi(k,t) = -\frac{1}{2}\mu|k|D(k,t) + \frac{1}{2}\mu|k| \int_{-\infty}^{t} W_{\mathrm{III}}(|k|c_s(t-t')) \dot{D}(k,t') \, \mathrm{d}t', \tag{31}$$

with $\Phi(k,t)$, $D(k,t)$ and $\dot{D}(k,t)$ being respectively the Fourier transform pairs of $\phi(x,t)$, $\delta(x,t)$ and $V(x,t)$. The mode III elastodynamic kernel $W_{\mathrm{III}}$ is defined in Morrissey and Geubelle[32] as a function of the Bessel

function of the first kind $J_1$:

$$W_{\mathrm{III}}(T) = \int_{T}^{\infty} \frac{J_1(\zeta)}{\zeta} d\zeta. \tag{32}$$

The transverse problem accounts for the thermo-hydro-mechanical shear response of a fluid-saturated fault gouge. Taking advantage of the time and length scale separation between the $x$ and $y$ directions existing in the problem of interest, heat and fluid flows through the granular material are only solved along the $y$ direction. Next, the assumption of fluid flow dominated by viscosity (Darcy's law with hydraulic diffusivity $\alpha_{\mathrm{hy}}$) and heat transfer by conduction (Fourier's law with thermal diffusivity $\alpha_{\mathrm{th}}$) are invoked to obtain a coupled set of diffusion equations that describes change of temperature $T$ caused by frictional shear and the associated change in fluid pressure $p$ due to thermal expansion of the porous medium. The *thermal pressurisation* equations read[2]:

$$\begin{aligned} \frac{\partial T}{\partial t} &= \frac{\tau \dot{\gamma}}{\rho c} + \alpha_{\mathrm{th}} \frac{\partial^2 T}{\partial y^2}, \\ \frac{\partial p}{\partial t} &= \Lambda \frac{\partial T}{\partial t} + \alpha_{\mathrm{hy}} \frac{\partial^2 p}{\partial y^2}. \end{aligned} \tag{33}$$

In the equations above, $\rho$ and $c$ are respectively the density and specific heat capacity of the gouge and $\Lambda$ is a mean-field parameter that describes the change of pore fluid pressure caused by an increase in temperature in the gouge. $\tau$ and $\dot{\gamma}$ are respectively the shear stress and strain rate in the gouge. Following Rice et al. [21], Platt et al. [22], we assume that the gouge follows a weak rate-hardening behaviour:

$$\tau = f_{\mathrm{rsf}}(\dot{\gamma})(\sigma_n - p) = (f_0 + A \ln(\dot{\gamma}/\dot{\gamma}_0))(\sigma_n - p), \tag{34}$$

defined by the friction parameters $f_0$, $A$ and $\dot{\gamma}_0$ and the normal stress $\sigma_n$. By analogy to rate-and-state friction, $A$ controls the 'direct' strengthening response of the gouge whereas the 'long term' evolution of the shear stress is given by the thermo-hydro-mechanical response of the gouge through the evolution of $p(y, t)$.

Throughout the rupture, the separation of scales between the longitudinal and transverse direction allows us to neglect (1) fluid and heat flows in the $x$ direction and (2) inertial effects through the thickness of the gouge. Consequently, the shear stress is invariant across the gouge and corresponds to

$$\tau(x, -h/2 \leq y \leq h/2, t) \equiv \tau(x, t). \tag{35}$$

Conversely, the strain rate varies across the gouge and is related to the macroscopic slip velocity via the following integration:

$$\int_{-h/2}^{h/2} \dot{\gamma} \, \mathrm{d}y = \int_{-h/2}^{h/2} \frac{\partial \dot{u}}{\partial y} \, \mathrm{d}y = \dot{u}(x, y = h/2, t) - \dot{u}(x, y = -h/2, t) \equiv V(x, t). \tag{36}$$

The stress (35) and kinematic (36) conditions above allow for coupling the longitudinal and transverse problems.

## Dimensional analysis

Defining $V_c = 2c_s \tau_b/\mu$, $t_c = \delta_c/V_c$ and $L_c = c_s t_c$, the elastodynamic equation (14) can be rewritten in dimensionless form as

$$\tilde{\tau}(\tilde{x}, \tilde{t}) = 1 - \tilde{V}(\tilde{x}, \tilde{t}) + \tilde{\phi}(\tilde{x}, \tilde{t}), \tag{37}$$

with $\tilde{\tau} = \tau/\tau_b$, $\tilde{x} = x/L_c$, $\tilde{t} = t/t_c$, $\tilde{V} = V/V_c$ and

$$\tilde{\Phi}(\tilde{k}, \tilde{t}) = -|\tilde{k}| \tilde{D}(\tilde{k}, \tilde{t}) + |\tilde{k}| \int_{-\infty}^{\tilde{t}} W_{\mathrm{III}}(|\tilde{k}|(\tilde{t} - \tilde{t}')) \dot{\tilde{D}}(\tilde{k}, \tilde{t}') \, \mathrm{d}\tilde{t}'. \tag{38}$$

**Table 1 | Table of dimensionless variables**

| Quantities | Variables | Definition |
|---|---|---|
| Pressure | $\tilde{p}$ | $(p - p_0)/(\sigma - p_0)$ |
| Temperature | $\tilde{T}$ | $\Lambda(T - T_0)/(\sigma - p_0)$ |
| Distance across gouge | $\tilde{y}$ | $y/h$ |
| Shear stress | $\tilde{\tau}$ | $\tau/\tau_b$ |
| Slip velocity | $\tilde{v} = V/V_c$ | $V \times \mu/(2c_s\tau_b)$ |
| Slip | $\tilde{\delta} = \delta/\delta_c$ | $\delta \times \Lambda f_c/(\rho ch)$ |
| Distance along fault | $\tilde{x} = x/L_c$ | $x \times (2\tau_b)/(\mu\delta_c)$ |
| Time | $\tilde{t} = t/t_c$ | $t \times (2c_s\tau_b)/(\mu\delta_c)$ |
| Strain rate | $\dot{\tilde{\gamma}} = \dot{\gamma}/\dot{\gamma}_c$ | $\dot{\gamma} \times (\mu h)/(2c_s\tau_b)$ |

The characteristic slip distance $\delta_c$ will be set later by the rheology of the interface. Defining $\dot{\gamma}_c = V_c/h$ and $f_c = f_{rsf}(\dot{\gamma}_c)$, the effective friction relationship can be rewritten as

$$\tilde{\tau} = \left(1 + \frac{A}{f_c}\ln(\dot{\gamma}/\dot{\gamma}_c)\right)\left(1 - \frac{p - p_0}{\sigma_n - p_0}\right)\frac{f_c(\sigma_n - p_0)}{\tau_b}$$
$$\equiv \left(1 + \tilde{z}^{-1}\ln(\dot{\tilde{\gamma}})\right)(1 - \tilde{p})\tilde{\eta}^{-1} \tag{39}$$

In the equation above, $\tilde{z} + \ln(\dot{\tilde{\gamma}})$ is further approximated as $\text{arcsinh}(e^{\tilde{z}}\dot{\tilde{\gamma}}/2)$ in order to rewrite the constitutive relationship as

$$\tilde{\tau} = \frac{1 - \tilde{p}}{\tilde{z}\tilde{\eta}}\text{arcsinh}\left(\frac{\dot{\tilde{\gamma}}}{2}e^{\tilde{z}}\right), \tag{40}$$

which has the advantage of regularising the stick-to-slip transition as $\dot{\tilde{\gamma}} \to 0$. In dimensionless form, the kinematic condition (36) writes then

$$\tilde{V}(\tilde{x}, \tilde{t}) = \int_{-1/2}^{1/2}\dot{\tilde{\gamma}}(\tilde{x}, \tilde{y}, \tilde{t})\,d\tilde{y} = \int_{-1/2}^{1/2}2e^{-\tilde{z}}\sinh\left(\frac{\tilde{z}\tilde{\eta}\tilde{\tau}(\tilde{x}, \tilde{t})}{1 - \tilde{p}(\tilde{x}, \tilde{y}, \tilde{t})}\right)d\tilde{y}. \tag{41}$$

The set of diffusion equations governing thermal pressurisation (33) can then be written in the following dimensionless form

$$\frac{\partial \tilde{T}}{\partial \tilde{t}} = \tilde{\eta}\tilde{\tau}\dot{\tilde{\gamma}} + \tilde{\alpha}_{th}\frac{\partial^2\tilde{T}}{\partial\tilde{y}^2},$$
$$\frac{\partial \tilde{p}}{\partial \tilde{t}} = \frac{\partial \tilde{T}}{\partial \tilde{t}} + \tilde{\alpha}_{hy}\frac{\partial^2\tilde{p}}{\partial\tilde{y}^2}, \tag{42}$$

where we used the following definition for the characteristic slip

$$\delta_c = \frac{\rho ch}{\Lambda f_c}. \tag{43}$$

Remarkably, in this dimensionless framework summarised in Table 1, the behaviour of the coupled system is controlled only by the following four dimensionless quantities:

$$\begin{cases} \tilde{\alpha}_{th} = \alpha_{th}\dfrac{t_c}{h^2} = \alpha_{th}\dfrac{\mu\rho c}{2c_s\tau_b f_c\Lambda h}, \\[2mm] \tilde{\alpha}_{hy} = \alpha_{hy}\dfrac{t_c}{h^2} = \alpha_{hy}\dfrac{\mu\rho c}{2c_s\tau_b f_c\Lambda h}, \\[2mm] \tilde{z} = \dfrac{f_c}{A}, \\[2mm] \tilde{\eta} = \dfrac{\tau_b}{f_c(\sigma_n - p_0)}, \end{cases} \tag{44}$$

recalling that $f_c = f_{rsf}\left(\dot{\gamma} = \frac{2c_s\tau_b}{h\mu}\right)$. The reference set of parameters used for the simulations are representative to a fault zone at 7 km depth[22] and corresponds to $\tilde{\alpha}_{th} = 0.0061$, $\tilde{\alpha}_{hy} = 0.075$, $\tilde{z} = 15$ and $\tilde{\eta} = 0.64$.

**Numerical methods**

The two diffusion Equations (42) solved in the transverse direction can be generically written as

$$\frac{\partial\tilde{\Psi}}{\partial\tilde{t}} = \tilde{S} + \tilde{D}\frac{\partial^2\tilde{\Psi}}{\partial\tilde{y}^2}, \tag{45}$$

and are numerically integrated using Crank-Nicholson method with an explicit source term:

$$\frac{\tilde{\Psi}_j^{m+1} - \tilde{\Psi}_j^m}{\Delta\tilde{t}} = \tilde{S}_j^m$$
$$+ \frac{\tilde{D}}{\Delta\tilde{y}_{j+1} + \Delta\tilde{y}_j}\left(\frac{\tilde{\Psi}_{j+1}^{m+1} - \tilde{\Psi}_j^{m+1}}{\Delta\tilde{y}_{j+1}} + \frac{\tilde{\Psi}_{j+1}^m - \tilde{\Psi}_j^m}{\Delta\tilde{y}_{j+1}} - \frac{\tilde{\Psi}_j^{m+1} - \tilde{\Psi}_{j-1}^{m+1}}{\Delta\tilde{y}_j} - \frac{\tilde{\Psi}_j^m - \tilde{\Psi}_{j-1}^m}{\Delta\tilde{y}_j}\right),$$

with $\tilde{t} = m\Delta\tilde{t}$ and an irregular grid along the $\tilde{y}$ direction with a spacing $\Delta\tilde{y}_j$ between the node $j$ and $j - 1$. Regrouping the unknowns at the next time step on the left-hand side, the equation above can be expressed as the following linear system:

$$\tilde{\Psi}_j^{m+1} = (\mathbf{P}_{lj})^{-1}(\tilde{S}_l^m + \mathbf{Q}_{ln}\tilde{\Psi}_n^m). \tag{46}$$

The matrix $\mathbf{P}_{lj}$ is tridiagonal and is efficiently inverted using Thomas algorithm.

The elastodynamics along the fault is solved through Equations (14) and (31). Along a longitudinal grid of equally spaced sampling points $\tilde{x}_i$, the non-local elastodynamic contribution is then integrated as

$$\tilde{\Phi}_k^m = -k\tilde{q}_0\tilde{\Omega}_k^m + k\tilde{q}_0\sum_{n=0}^m W_{III,k}^{m-n}\dot{\tilde{\Omega}}_k^n\Delta\tilde{t}, \tag{47}$$

with $W_{III,k}^m = W_{III}(k\tilde{q}_0 m\Delta\tilde{t})$ being the convolution kernel whose integration according to Equation (32) is computed through a polynomial approximation[32,49]. $\tilde{\Omega}_k$, $\dot{\tilde{\Omega}}_k$ and $\tilde{\Phi}_k$ are the discrete Fourier mode of respectively $\tilde{\delta}_i$, $\tilde{V}_i$ and $\tilde{\phi}_i$. The discrete Fourier transform introduces a longitudinal periodic boundary condition with period $\tilde{X}$ such that the fundamental wave number $\tilde{q}_0 = 2\pi/\tilde{X}$. Combining Equations (37) and (41), the continuity of shear stress through the gouge at time $\tilde{t} = m\Delta\tilde{t}$ requires that:

$$\zeta(\tilde{\tau}_i^m) \equiv \tilde{\tau}_i^m - 1 + \sum_{j|y_j\in[-\frac{1}{2},\frac{1}{2}]}2e^{-\tilde{z}}\sinh\left(\frac{\tilde{\eta}\tilde{z}\tilde{\tau}_i^m}{1 - \tilde{p}_{i,j}^m}\right)\Delta\tilde{y}_j - \tilde{\phi}_i^m = 0. \tag{48}$$

$\zeta$ is a monotonic and differentiable function of $\tilde{\tau}_i^m$ whose root is found iteratively using a Newton-Raphson scheme. Finally, the slip velocity is integrated from the solution $\tilde{\tau}_i^m$ as

$$\tilde{V}_i^m = \sum_{j|y_j\in[\frac{1}{2},\frac{1}{2}]}2e^{-\tilde{z}}\sinh\left(\frac{\tilde{\eta}\tilde{z}\tilde{\tau}_i^m}{1 - \tilde{p}_{i,j}^m}\right)\Delta\tilde{y}_j. \tag{49}$$

At time $\tilde{t} = m\Delta\tilde{t}$, the coupled numerical scheme is integrated in time by the following predictor-corrector scheme:
1. Compute $\tilde{p}_j^{m+1}$ and $\tilde{T}_j^{m+1}$ following Crank-Nicholson integration scheme (Equation (46))
2. Integrate predictor slip: $\tilde{\delta}_i^* = \tilde{\delta}_i^m + \Delta\tilde{t}\,\tilde{V}_i^m$
3. Using the predictor slip, compute the predictor velocity $\tilde{V}_i^*$ by solving Equations ((47)-(48)-(49))

4.  Correct slip integration: $\tilde{\delta}_i^{m+1} = \tilde{\delta}_i^m + \Delta\tilde{t}(\tilde{V}_i^m + \tilde{V}_i^*)/2$
5.  Integrate the non-local dynamic contribution $\tilde{\phi}_i^{m+1}$ using Equation (47)
6.  Compute frictional stress $\tilde{\tau}_i^{m+1}$ using Equation (48)
7.  Compute slip velocity $\tilde{V}_i^{m+1}$ using Equation (49)

### Initial and boundary conditions

The system of interest consists of a shear zone lying between two semi-infinite linear elastic wall rocks as shown in Fig. 1. The system is initially at rest under homogeneous pressure $p_0$, temperature $T_0$, shear $\tau_b$ and normal $\sigma_n$ stress. At time $t = 0$, the rupture event is nucleated by introducing a Gaussian perturbation of the pore pressure at the centre of the fault such that

$$\tilde{p}(\tilde{x}, \tilde{y}) = \tilde{\Pi}\exp\left(-\frac{\tilde{x}^2}{2\tilde{\xi}^2}\right), \quad \text{for } \tilde{y} \in [-1/2, 1/2].  \quad (50)$$

The amplitude and standard deviation are typically set respectively to $\tilde{\Pi} = 0.6$ and $\tilde{\xi} = 5\%\tilde{X}$. The system is assumed to be infinite in the transverse directions, whereas periodic boundary conditions in the longitudinal direction (with period $\tilde{X}$) are assumed to compute the discrete Fourier modes $\tilde{\Omega}_k$, $\dot{\tilde{\Omega}}_k$ and $\tilde{\Phi}_k$ used in Equation (47).

### Stability and convergence

Crank-Nicholson time integration defined in Equation (46) is unconditionally stable and the definition of the stable time step is set by time integration of the elastodynamic model following the Courant-Friedrichs-Lewy condition:

$$\Delta\tilde{t} = \beta_{\text{CFL}}\Delta\tilde{x}.  \quad (51)$$

A value of $\beta_{\text{CFL}} = 0.1$ has been used in the simulations reported in this paper. The longitudinal direction is regularly discretized with $N_x$ sampling points, such that $\Delta\tilde{x} = \tilde{X}/N_x$. In the transverse direction, the gouge thickness is regularly sampled by $N_y + 1$ points, such that $\Delta\tilde{y} = 1/N_y$. To capture the exponential decay of pressure and temperature fields outside the gouge layer, the numerical domain symmetrically stretches into the surrounding wall rock, which is sampled with $N_y$ additional points that are logarithmically-spaced such that $\Delta\tilde{y}_{j=1} = \Delta\tilde{y}_{j=2N_y} = 1$. Background pore pressure and temperature $\tilde{p} = \tilde{T} = 0$ are imposed at the two edges ($y_{j=0}$ and $y_{j=2N_y}$) of the numerical grid, whose distance from the centre of the gouge should exceed the diffusion length scale:

$$\tilde{y}_{j=2N_y} > 2\sqrt{\tilde{\alpha}N_t\Delta\tilde{t}},  \quad (52)$$

with $\tilde{\alpha} = \sqrt{\tilde{\alpha}_{\text{hy}}^2 + \tilde{\alpha}_{\text{th}}^2}$ and $N_t$ being the total number of time steps. Typical simulations use $N_x = 4096$ and $N_y = 200$, which are sufficient to reach numerical convergence.

### Thermal pressurisation neglecting strain localisation

When the possibility for strain-localisation is neglected, strain rate keeps the same transverse profile over time:

$$\dot{\tilde{\gamma}}(\tilde{x}, \tilde{y}, \tilde{t}) = \tilde{V}(\tilde{x}, \tilde{t})\mathcal{F}(\tilde{y}).  \quad (53)$$

In the literature, the strain rate through the gouge is often assumed to follow a Gaussian profile and we take the following profile

$$\mathcal{F}(\tilde{y}) = \frac{4}{\pi}\exp(-\pi\tilde{y}^2),  \quad (54)$$

in order to satisfy the kinematic condition (36). Neglecting strain localisation also implies that the rheology only depends on the conditions at the centre of the gouge layer such that the friction law (40) becomes

$$\begin{aligned}\tilde{\tau}(\tilde{x}, \tilde{t}) &= \frac{1 - \tilde{p}(\tilde{x}, \tilde{y} = 0, \tilde{t})}{\tilde{z}\tilde{\eta}}\text{arcsinh}\left(\frac{\dot{\tilde{\gamma}}(\tilde{x}, \tilde{y} = 0, \tilde{t})}{2}e^{\tilde{z}}\right) \\ &= \frac{1 - \tilde{p}_0(\tilde{x}, \tilde{t})}{\tilde{z}\tilde{\eta}}\text{arcsinh}\left(\frac{2\tilde{V}(\tilde{x}, \tilde{t})}{\pi}e^{\tilde{z}}\right)\end{aligned}  \quad (55)$$

and now explicitly depends on the slip rate.

## Data availability

The code used to produce the data of this paper is freely accessible (see details in the 'Code availability' Section).

## Code availability

The code and scripts used to run the simulations, analyse data and produce the figures of this article have been deposited in the Zenodo database https://doi.org/10.5281/zenodo.14259186.

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

## Acknowledgements

This project has received funding from the European Research Council under the European Union 's Horizon 2020 research and innovation program (project RockDEaF, grant agreement No. 804685 to N.B.), from the UK Natural Environment Research Council (project NE/S000852/1 to N.B.) and from the Research Council of Norway (project UNLOC, grant agreement No. 345008 to F.B.). The authors are thankful to Massimo Cocco, Dmitry Garagash and John Platt for insightful comments at various stages of this work.

## Author contributions

F.B. and N.B. contributed to: Funding acquisition, Conceptualisation, Software, Investigation, Formal analysis, Writing and Visualisation.

## Competing interests

The authors declare no competing interests.
