## [Transparent Peer Review file · Nature Communications]

Shear localisation controls the dynamics of earthquakes

Corresponding Author: Dr Fabian Barras

Version 1:

Reviewer comments:

Reviewer #1

(Remarks to the Author)

Please find my review in the pdf attachment below.

Reviewer #2

(Remarks to the Author)

This is an extremely interesting paper presenting inspiring and original results on a key issue in earthquake mechanics, namely what controls earthquake dynamics during rupture propagation. I think that this work can potentially be of significance to the field and in related fields, such as fracture mechanics, after some (likely) substantial revisions. I am not completely convinced of the interpretation of the results to support conclusions and claims. It is not a matter of providing additional evidence, rather to balance the interpretations reconsidering some conclusions. The methodological contributions are rigorous and convincing; they are presented with excellent standards and a suitable approach. The methods are also presented with a sufficient detail ensuring reproducibility and a full understanding of the theoretical framework. I will present in the following my doubts on the provided interpretations and conclusions, hoping to be helpful and to clearly provide suggestions useful for the necessary revision of the manuscript.

The authors propose the following conclusions:

- Earthquakes are likely to be systematically associated to extreme strain localization
- Strain localization controls dynamic weakening and creates an abrupt drop of shear stress promoting the propagation of classical dynamic ruptures (i.e., according to LEFM)
- The sudden localization of shear strain within a shear zone leads to the emergence of classical cracks driven by a constant fracture energy.

I am personally convinced that strain localization plays a dominant role during faulting and earthquakes. It is strain localization that enables the creation of the fault core and the principal slipping zones (PSZs). This implies that strain localization is acting in different stages of the earthquake cycle, and it is not limited to the coseismic "additional" localization proposed in this study. Geological faults are characterized by fault cores with multiple PSZs. This implies that the strain localization process during individual earthquakes can be associated with different PSZs, likely suggesting heterogeneities of gouge materials which are not accounted by the simple theoretical definition of pressurized gouge used in this paper (i.e., constant shear stress in the fault-normal direction). This should be accounted by the authors by smoothing a bit the ambition to extrapolate their theoretical results to all earthquakes and all faults.

I think that expressing shear stress evolution as a function of strain and strain rate represents a groundbreaking achievement in earthquake mechanics. Several papers in the literature attempted to tackle this challenge (see Beeler et al, JGR 1996), but a full representation of stress as a function of strain and strain rate is still missing. The authors try to address this issue but, according to my understanding, in this paper the shear stress is (solely) related to strain rate through sliding velocity. In other words, the dependence of shear stress on strain rate is not explicit. I would suggest clarifying the shear stress dependence on sliding velocity, since the latter controls strain localization, elastodynamics and thermal pressurization (through the heat source). It is unclear to me why strain localization dominates over others, as I will explain in the following. My main concern about the interpretations applies to the description of the coupled dynamic system used to describe earthquake propagation. Immediately after nucleation along the creeping fault, the rupture starts to propagate driven by elastodynamics (stress depends on sliding velocity as stated in equation 12), which is expected to evolve according to

dynamic weakening. The adopted model includes thermal pressurization, which increases dynamic weakening, potentially leading to extreme dynamic weakening. Thermal pressurization requires a heat source and fluids; thus, it is activated by the initial dynamic weakening and slip acceleration. As the rupture propagates and slip velocity increases, strain is further localized within the fault zone because of the dependence of strain rate on slip velocity. Is this coseismic strain localization the effect or the cause of dynamic propagation? The authors interpret the result proposing strain localization as the cause, but I am not fully convinced by their reasoning. Because of its definition (strain is assumed to scale with slip and thickness), it is not surprising that increasing slip velocity increases strain rate, but the engine should be the elastodynamics and thermal pressurization that are the main drivers of dynamic weakening. In other words, I agree with the results shown in Figure 4, but they demonstrate the relation between slip velocity and strain rate, coherently with the adopted theoretical definition. I don't see these results as evidence of the dominant role of strain localization. What is the evidence corroborating that dynamic weakening is driven by strain localization and not by thermal pressurization?

The shear stress evolution with position (figures 2 and 4) and slip (figure 3) is quite interesting and, in my opinion, it deserves a careful interpretation. Looking at the slip weakening curves shown in figure 3 the shear stress initially raises to its peak value and the subsequent stress breakdown is divided in different stages with different slopes (stress gradient with slip). Can the authors identify and distinguish the contributions of thermal pressurization and strain localization during the dynamic breakdown stage? Can they prove that the abrupt weakening (shown in figures 3 and 4) is due to strain localization and not to thermal pressurization?

The association of slip velocity evolution with strain rate shown in Figure 4 does not corroborate, in my opinion, the dominant role of strain localization in controlling dynamic weakening. First, because this is the result of the analytical dependence of strain and slip rates. Second, because dynamic weakening inferred from stress evolution is very different from LFM predictions. The authors just compare inferred shear stress and residual stress resulting from LFM (or the initial raise before peak stress). Third, because peak slip velocity is reached at the end of the dynamic weakening stage. Dynamic weakening is associated with slip acceleration (points B and C in Figure 4), when strain is not yet fully localized. Strain localization is reached at the peak slip velocity (C in Figure 4), when dynamic breakdown is completed. Why is strain localization is considered the driver of dynamic weakening?

Interpreting figure 3, the authors state that predictions of shear stress at the tip $\tau_{sp}(\delta, V_{tip})$ provide a good estimate of the residual shear stress reached after strain localization. I don't see such a corroborating fit and, more important, it is not clear to me why thermal pressurization should provide nearly constant or smoothly decreasing residual stress with slip (again, the authors should discuss why strain localization – interpreted as the driver of dynamic weakening, should overcome thermal pressurization) as shown in figures 3 and 4. I am also a bit surprised that, while slip velocity displays the expected behavior around the tip, the shear stress does not show any evident peak (as expected for a regularized singular field); rather it displays a very modest strength excess. This is not coherent with LFM predictions, in my opinion. It might be also useful to compute the shear stress evolution inferred from thermal pressurization only (without strain localization) using it for comparison (also discussing the parameters' ranges in which thermal pressurization is weak or strong). The authors should better explain this behavior and discuss it to support their claims. Are we sure that this behavior is not due to the peculiar range of model parameter values used to simulate thermal pressurization?

The interpretation of fracture energy and its scaling with slip during rupture propagation does not completely convince me. The authors claim that the good agreement with LFM predictions shown in Figure 4 confirms the crack-like nature of the simulated rupture process near the tip of the slipping patch. They interpret this (debatable, see above) result to justify the computation of fracture energy from stress intensity factor (equation 15), governing the propagation of the dynamic rupture according to LFM and small-scale yielding approximation (a first scale separation, see below). They therefore compute $G(\delta)$ (as defined by Abercrombie and Rice, 2005, equation 16) considering it the breakdown work in systems in which "frictional weakening" does not necessarily reach a well-defined residual value. However, looking at figures 3 and 4, the minimum shear stress to compute breakdown work following its definition (Tinti et al., 2005; Cocco et al., 2005) can be identified. The authors might also compute breakdown work. This is not a major issue and I understand that the authors implicitly distinguish fracture energy (tip dissipation) from frictional weakening. What is wondering me is which portion of shear stress evolution with slip during dynamic weakening corresponds to G_c from LFM. The authors try to answer to this question interpreting figure 5, showing the scaling of breakdown energy (or breakdown work) with slip in different positions during the rupture propagation. Simulations with strong strain localization show a plateau corresponding to the G_c value. Can the authors interpret the range of slip values for which EBD is equal to G_c and the plateau appears? To which spatial positions these slip values correspond? Why there is only a given range of slip values for which EBD is equal to G_c ? What differs from smaller or larger slip values? Furthermore, looking at the shear stress evolution with slip, it is not surprising that breakdown work differs from fracture energy, at least because there is an initial hardening with non-negligible slip (i.e., there is mechanical work spent during strength excess).

The result shown in figure 5 is very interesting. To me, this confirms that breakdown work scales with slip during the dynamic propagation (as previously stated for pulse-like ruptures, Tinti et al., 2005; Rice et al., 2005; Cocco et al., 2006, among several others). If breakdown work is associated with frictional weakening, while constant fracture energy is associated with near-tip dissipation (I personally do not agree with this distinction, but this does not matter here), why they become equal only in a given slip range during rupture propagation? This result requires in my opinion a more comprehensive discussion. I am not asking to discuss the scaling of global estimates of fracture energy G' or breakdown work with total slip (i.e., a single value for each earthquake) as discussed by Cocco et al. (2023), because here we are focusing on the fracture energy and breakdown work values during dynamic propagation in different fault positions. Figure 5 clearly shows that a plateau appears when strain rate is localized, which means high slip rates. Can the authors discuss the dynamic propagation of a weakly localized rupture? Which slip velocity evolution is inferred in this case?

The discussion of scale separation also requires some clarifications. The authors state that strain localization provides a clear separation between cohesive zone and the interior of the slipping patch, hence justifying the small-scale yielding hypothesis. This needs in my opinion clarifications. For classic LFM predictions the scale separation concerns the zone of inelastic yielding near the crack tip and the large-scale elastic response of the bulk. Here, there is a scale separation between processes affecting strain localization within the thickness of shearing layer and along the crack length (transverse

and longitudinal directions, as explained in the Appendix). Can the authors better explain why these different scales should be all coherent with the small-scale yielding hypothesis? The scale separation for LEFM is associated with a nearly constant residual stress independent of slip. Why strain localization should imply a nearly constant residual stress counterbalancing thermal pressurization, which on the contrary would imply a continuously decreasing shear stress with slip? The authors mention the cohesive zone only at the very end of the paper. According to elastodynamics, the size of the cohesive zone depends on slip velocity evolution. Once again, I remain with the doubt of which is the cause, and which is the effect. Everything turns around the evolution of slip velocity, which occurs during dynamic weakening. It would be very useful to interpret the breakdown stage by analyzing the stress evolution, and not just the slip velocity profile.

In conclusion, I am convinced that the paper addresses a relevant issue in earthquake mechanics. The results might have a relevant impact for future investigations if interpretations are better explained and corroborated. Reading the paper, I was left with the impression that the reader is led to follow the interpretation preferred by the authors without having a complete understanding of the implications of this study. Perhaps this is just the narrative approach taken, but I remain of the opinion that the results should be discussed in a broader and more comprehensive way. For these reasons I recommend major revisions to strengthen the interpretation of the results justifying conclusions and claims.

Version 2:

Reviewer comments:

Reviewer #1

(Remarks to the Author)

Reviewer #2

(Remarks to the Author)

The authors did an excellent job in revising the manuscript, addressing my previous comments and indications as well as responding to those of the other reviewer. I am convinced that this is an extremely interesting paper presenting inspiring and original results on key issues in earthquake mechanics, still debated in the community. The authors revised the manuscript in an efficient way, making the paper more accessible to readership, emphasizing the original results of their study making them more accessible. I believe that the paper is ready to be accepted for publication, but I would suggest a further check of the text and some minor improvement of the interpretations.

Here some indications that I hope will be useful to further improve the manuscript.

1. Comparison with classic dynamic crack propagation. The comparison with classic solutions from dynamic fracture mechanics is central to the paper. The authors use different terms to mention solutions from dynamic fracture mechanics, which might be misleading. They use "classical cracks driven by constant fracture energy" in the abstract (which is appropriate). Therefore, you mention LEFM and singular solutions at the tip of a dynamic fracture (rows 260-263 of the manuscript with track changes). Therefore, you mention "dynamic fracture solution (row 268) and you refer to the process zone (row 272) process region (row 289). You refer to the crack-like nature of rupture dynamics (row 304). You cite the cohesive zone (row 341 and 444). You corrected the reference to LEFM in favor of dynamic fracture theory (row 365) and crack like rupture (row 427), but you maintain the predictions of LEFM in row 393. I would suggest referring to dynamic fracture theory (not LEFM, if not necessary) and cohesive zone. This for several reasons. First, your simulations do not include any singular field at the crack tip; therefore, a classic dynamic fracture theory with a cohesive zone (Palmer and Rice, 1973) well describe the comparison. Second, because you have a peak shear stress and a breakdown stress drop, and slip velocity is finite (not singular). In any case, I would recommend homogenizing the terminology.

2. The boundary integral formulation (equation 14). I am not sure it is appropriate to call the contribution of dynamic interactions ($\phi(x,t)$ in equation 14) "non-local dynamic contribution", because this open the question of the cohesive zone size and the slip velocity decrease to the residual value. I understand that you want to distinguish local dissipation, but you cannot exclude that your dual-scale model implies a cohesive zone size which makes relevant (i.e., non-negligible) the contribution from dynamic interactions in a region around the rupture front involving local and/or nearly local slipping points.

3. Separation of scales. In rows 211-214 you support the separation of scales making a reference to the parameterization of the problem (diffusive processes at the scale of the strain rate localization are smaller than the few meters-long distance between grid points). Is the scale separation depending on grid size (problem parameterization)? I don't think this is the case, but this sentence is misleading. I would suggest revising it.

4. Interpretations of dynamic rupture solutions. There are still some interpretations that require, in my opinion, some check and clarification. I mention here some issues that I noticed but I would suggest a careful check by the authors.

- In figure 3 the residual stress does not seem to be constant and independent of slip. However, commenting Figure 4, you

refer to a constant residual stress (rows 292-294).

- Your fracture energy estimates G_c are associated to a portion of the stress breakdown stage (and stress drop). Can you show this portion on the slip weakening curves (shear stress versus slip)? The dynamic weakening of shear stress shows a change of slope associated with the peak slip velocity, which should correspond to the transition between the edge-localized dissipation (according to your interpretation) and dissipation driven by thermal pressurization. Most of the breakdown stress drop occurs over the Δ_c , which is associated with the strength reduction. I understand that the strain localization causes a dramatic reduction of the stress-bearing capacity, which promotes the subsequent strength reduction driven by thermal pressurization, but I would recommend to better explain this important aspect. I find this discussion still unclear in the manuscript.
- Why the slip acceleration to peak slip velocity involves the same amount of slip and a nearly similar shear stress evolution around its peak? Your G_c are estimated through the comparison with elastic fracture mechanics, resulting from the comparison with simulations and estimates of elastic stress intensity factor (equation 17). Why this should work for a gouge material which is not elastic? Because of high strain rates caused by localization. This is a minor aspect for the revision, but it is not clear to me.

Minor corrections

- The legend of Figure 5 is wrong: the blue lines refer to strong strain localization
- The authors refer to equation (4.1) (rows: 331, 335, 402, at least), which is not included in the text. I guess this is equation (8) or Section 4.1? I am not sure. Please check.
- Substitute rupture with fracture at row 391. I guess you refer to fracture energy.

I hope these comments will be useful. I encourage you to make a further check and spend some more efforts to make your paper easily and fully accessible to readers. It is an extremely inspiring and timely contribution.

Yours Sincerely
Massimo Cocco

Dear anonymous Reviewers,

We first want to thank you for the thorough reviews of the paper and valuable feedback that helped us improving the manuscript. We appreciate the positive evaluation of our work and the critical description of the aspects that needed improvement and clarification. To address the review, we did a major revision of the paper that includes the implementation of a second type of model identical to the benchmark approach used in the literature to simulate seismic ruptures with thermal pressurisation. The systematic comparison with the dual-scale model proposed in our work clearly highlights the novelty of our approach and strengthens the message of the paper. Two main aspects have been raised by both reviewers and lead to major changes in the manuscript.

- *The need for a better mechanistic description of the model and comparison with the 1D kinematic model of [1].* To address this important point, we have substantially rewritten Section 3 of the paper. At the micro-scale, we now provide a detailed mechanistic description of the two regimes, adiabatic undrained and slip-on-a-plane. The new version also presents the analytical expression of the weakening behavior predicted for the two regimes. At the macro-scale, we connect these two weakening behaviors to the breakdown energy of the fault. The revised version also explains why the applicability of 1D kinematic model to study the dynamics of seismic ruptures remains unclear and motivates the need for the coupled model developed in this work.
- *The "chicken and the egg" question.* This is a very relevant aspect that ties to the question of what controls the propagation of seismic rupture central to our manuscript. We fully agree that the process of strain localisation and thermal pressurisation are closely interlaced. Thermal pressurisation (TP) is at the heart of the feedback loop initiating localisation, as evidenced by the linear stability analysis of [2]. The key question raised and addressed in our paper is to identify the mechanism and physics that control the propagation of failure along the fault. To clarify this aspect, we chose to implement and run a second type of model, also based on thermal pressurisation. This model is a benchmark to simulate thermal pressurisation in the context of seismic ruptures and only differs from our dual-scale approach by removing the possibility of strain localisation. The model without localisation produces rupture dynamics different from the one observed in our dual-scale model, where no edge-localised dissipation can be isolated. The new version of Figure 5 evidences how strain localisation introduces a clear separation that does not exist in the model without localisation. The latter features a smooth transition between the two asymptotic regimes—from adiabatic undrained to slip-on-a-plane conditions—as described in the work of [3]. Building on this comparison, the revised manuscript explains how the strain localisation phenomenon leads to the emergence of crack-like dynamics in agreement with linear elastic fracture mechanics (LEFM) and the possibility to identify G_c , the energy that drives it. In short, strain localisation is an instability caused by TP that drives the propagation of rupture at much lower fracture energy and with a substantially different rupture dynamics than the one driven by TP only and is, thereby, of prime importance to understand the dynamics of earthquake ruptures.

We have addressed all the comments and you can find, hereafter, a detailed reply to each point raised by the reviewers. We have copied in blue font the original comments and appended our replies afterwards in black font with reference to the lines in the revised version of the manuscript where related changes have been made. To easily identify the changes made to the document, a *latexdiff* pdf is submitted together with this document.

Yours Faithfully,

Fabian Barras, Nicolas Brantut

Reviewer #1

The authors present a model of a planar fault under a mode III configuration (2D model) which account for the following physics:

- Elastodynamics of the solid material (which is thus assumed linear elastic) surrounding the fault,
- The “core” of the fault zone (the gouge) is modeled as a 1D fluid-saturated thermo-poro plastic material. This 1D model is in all points similar to the one of Rice et al. (2014), Platt et al. (2014). I will refer to it as the 1D-fault plane perpendicular model - or fault 1D gouge model hereafter. The gouge has a finite thickness and shear localization delocalization occurs within this initial gouge layer.
- Separability of scales is assumed between the “small” scale 1D-fault plane perpendicular model of the fault gouge and the macroscopic zero-thickness displacement discontinuity model of the rupture. The gouge model is coupled to the macroscopic model via the condition of homogeneous shear stress in the gouge and the definition of the macroscopic slip velocity as the integrated small-scale shear strain rate across the gouge.
- The fault gouge is assumed to exhibit a rate-hardening frictional rheology at critical state (i.e. without any inelastic porosity changes in response to strain). Therefore switching off the stabilization effect of dilation in the model of Rice et al. (2014), Platt et al. (2014). In this contribution, localization due to thermal-pressurization is therefore in sole competition with the rate-hardening frictional rheology.
- Homogeneous properties are assumed throughout.

The novelty of the paper is the coupling between a macroscopic equilibrium model (usual slip on a plane elastodynamics model of a fault) with the 1D finite thickness gouge model of Rice et al. (2014), Platt et al. (2014). The latter was solved under kinematically controlled conditions (imposing the strain rate) by Platt et al. (2014) and can thus be considered as a “local” although complex (time dependent etc.) constitutive behavior for the macroscopic problem. It is important to note that the coupling with macroscopic elastodynamics allow to solve for spontaneous rupture propagation. As the 1D gouge model strain localizes under some conditions usually met when Thermal Pressurization (TP) is active, it is not a surprise that one recovers macroscopic crack-like behaviors which are impacted by localization-delocalization due to the finite gouge thickness. Some of the conclusions of the paper therefore already lays in the solution of the 1D finite thickness gouge model under kinematic conditions (Platt et al. 2014).

To clarify this aspect, we have now included simulations of seismic rupture run with the benchmark model for thermal pressurisation that prevents localisation. As discussed above, the comparison clarifies the different dynamics caused by strain localisation in comparison to the dynamics of seismic rupture only driven by TP studied in previous works [3]. The important change caused by localisation could not be anticipated from the studies of [1,2] that do not include any dynamic coupling. We have now developed a new paragraph between line 130-151 to clarify the contributions known from the 1D work of [1,2] as well as the unresolved questions that we address in the paper.

In particular, the condition for a localization instability derived here is merely a re-expression of Rice et al. (2014) . The authors mention general “diffusive-like” effects (coupled to frictional rheologies) in section 2, but after focus solely on the classical thermal-pressurization (TP) phenomena: I would suggest to just focus on TP and state that the stability analysis is the one of Rice et al. (2014) (because this is what it is).

We fully acknowledge that the linear stability behind the origin of localisation instability is not an original contribution from our work. Our goal with Section 2 is to make the connection between different works scattered in the earth science, physics and engineering literature, which is often overlooked (as evidenced by the lack of cross references between the papers that we gather in our manuscript). The fact that the same instability arises from other rheologies is important and strongly suggest that the effect of localisation reported with TP in our manuscript can similarly affect a broader type of geological contexts and systems. For this reason, we decided to keep this section, which also has the merit to introduce the mechanistic feedback at play behind localisation. However, we have reshaped the beginning of the section (between lines 71-79) to make the reference to past work much clearer.

However, the coupled dynamics simulations presented in this paper are interesting in several aspects: 1) they provide a quantification of the effect of strain localization on the so-called ‘break-down’ work evolution with accumulated slip for such planar ruptures. (Although one should recognize that such evolution is mostly governed by the physics of TP).

As detailed in the introduction of the letter, the revision of the manuscript and the comparison with the new simulations clarify how localisation is indeed governed by the physics of TP, but the rupture dynamics driven by localisation is substantially different from seismic ruptures produced only by TP.

2) they highlight a relation between rupture speed and the minimum thickness of the localized strain rates in the gouge (even though the derived approximation eq.(17) is not particularly convincing when compared to simulations).

The aim of the theoretical prediction shown in Figure 6 is to provide a simple relationship connecting predictions at the micro-mechanical level (i.e. the results of [1, 2]) and at the macroscopic scale (i.e. dynamic fracture theory [4]). The agreement is not stunning, but considering the simplifications used to construct (17), it is reasonably good at predicting the reduction of the localisation thickness over several orders of magnitude, including for simulation with heterogeneous stress conditions. The new version better emphasizes and discuss the limitation and merit of this approximation between lines 295-303.

1 Section 2 - shear localization and faulting & section 3 - Model

As already mentioned, I am not a fan of presenting a “general” model coupling diffusive process with shear strain/stress when in fact the whole remaining paper focus solely on thermal pressurization (TP) and rate hardening friction. The ‘general’ linear stability analysis (detailed in appendix A) is nothing more than a mere recasting of previous works (Bai 1982, Rice et al. 2014). This must be clearly stated.

As detailed above, we fully agree that the linear stability is not a contribution from our work. (The papers of [5], [2], [6] were already cited in the first version). We have made this clearer in the new version by adding lines 71-79.

It would be best to have more discussion / precision on some of the assumptions of the model. Notably, the exact boundary conditions for T&p (I had to go in the numerical description in the appendix to fully understand them). It’s notably important as - from what I understand - the stability analysis assume a no-flux boundary conditions at the gouge boundary (so-called undrained /adiabatic limit), which is different than the numerical simulations which model diffusion in the surrounding medium perpendicular to the rupture. Basically, not all reader will be accustomed to the undrained/adiabatic and drained limits - which are key to understand the underlying effect of thermal-pressurization on fault slip.

Thank you for underlying this aspect. We have now clarified the mechanistic aspects and the boundary conditions relevant for the undrained adiabatic and slip-on-a-plane limits between lines 133-148. Both end-member solutions are now written and explained in the main text.

An even more important point worth discussing in more details is when does the “separability” of scales actually “holds” (with respect to accumulated slip versus gouge thickness etc.). A priori, the coupling of a macroscopic slip along a planar zero-thickness macroscopic interface with the local 1D gouge model necessarily “forces” localization to happen and as result a crack-like behavior necessarily emerge. This must be recognized when analyzing the “generality” of the results. This

is a good point that ties to the nucleation procedure. The proposed multi-scale model rests upon the separation of scale and assumes that the length scale of diffusion processes over the duration of the seismic rupture remains smaller than the separation between grid points of the elastodynamic domain. This assumption is relevant with the dynamic rupture stage studied in this paper. At nucleation, this assumption might not always hold—such as in the case of marginally pressurised fault zones. Future work can adapt our coupled model to study the impact of strain localisation at nucleation and the diversity of slip events that it can produce. This explanation has been added to the text at the end of section 3 between lines 183-190.

2 Section 4 - results

As already mentioned, some features of the results are directly “built-in” the response of the 1D gouge model. I believe the results presented therefore must be compared with the ones of Platt et al. (2014) in much more details (Platt et al. (2014) provides the solution under kinematic control). As such, one can also obtain an evolution of the breakdown work as function of accumulated slip from the results of Platt et al. (2014). These results of Platt et al. (2014) can be post-process with the slip-rate history obtained here numerically to obtain another estimation of the break down work & this must be plotted against the results obtained here - e.g in Fig. 5. One would expect close agreement in most cases.

We have already addressed this aspect through different changes discussed earlier in the reply. We are not sure if we understood correctly the reviewer’s suggestion: If we plug the history of slip rate observed in our simulation as boundary conditions of the 1D model of [1], we will—by construction of our model—obtained exactly the same weakening behaviour and breakdown work. The underlying question is how can the history of slip profile be known a priori without running the coupled model. In the paper, we answer this question by proposing an approximate description of the feedback between the micro-scale model of [1] and the rupture dynamics. We have now clarified the context of this prediction in lines 290-304.

From the main text, it is difficult to grasp easily what is varied between simulations. It took me a while to understand (back and forth between main text, appendices and some thinking). The dimensional analysis in the appendix C indicates that the problem depends on 4 dimensionless quantities, and that they are taken representative of a fault zone at 7km depth for the “strong localization” simulation in the main text. Another simulation (weak localization) is performed by increasing the dimensionless hydraulic diffusivity 4 times higher (which is known to “tone” down the effect of TP). So 2 points in the dimensionless parameter space is probed. This is fine as the results are sufficiently rich, but the way the results are plotted & analyzed render their understanding quite difficult (at least to me). Here is what I understood, and few remarks.

This point has been clarified by removing the simulation featuring weak localisation and replacing it by the results from the model preventing localisation (i.e., no localisation *at all*). The revised version states that the simulation studied is representative of the rupture dynamics observed with other set of parameters with our artificial nucleation procedure (lines 304-308). Studying the

variety of slip events produced by our dual-scale model is indeed an interesting question that can be investigated in a dedicated paper centered around the nucleation problem.

- Fig 2 highlights well the general evolution of the rupture as one would guess from the usual evolution of dynamic rupture and the results of Platt et al. (2014). Fig. 3 displays the quantities (macroscopic V and τ , micro-scale W_{loc}) as function of accumulated slip (macroscale quantities) at different location x/L_c along the 1D space. Lower x/L_c are positioned closer to the nucleation zone and therefore experience a rupture that has not completely 'developed'/'localized'. The rupture is accelerating while developing in all the presented results.

- Fig. 4 - I understand very well how the stress intensity, residual strength and tip position are fitted from the profile of the slip rate in the moving tip coordinate system (and then G_c estimated). Nevertheless I was a bit puzzled as to i) when the profile is taken during the simulation (toward the end I assume?) ii) if the fit was performed on more than one profile (i.e at more than one time during the simulation), if the resulting value of G_c was varying? I think that the profile is taken toward the end of the simulation, where the rupture has well developed toward a 'crack' like behavior away from the early-stage nucleation effect - but I could not be 100% sure from the manuscript. I imagine a sort of "goodness" of fit criteria for the LEFM model to apply can be reported as function of time to define when a well defined G_c emerges.

We have now clarified the position where the fit are made and added additional figures (Fig. 7-8) in Appendix that present the same plot for other locations along the fault and confirm that the value of G_c inverted from the fit is the same for all the points outside the nucleation region. These figures also provide the root-mean-square-error of the fitting procedure.

- Fig. 5 the breakdown work is reported for the same x/L_c position than Fig. 3 (this could be label in the Fig to help the readers). So for the strong localization, dark blue means position further away from the nucleation patch (and less "perturbed" by the initial phase of the rupture). A clear plateau at constant G_c emerges. What is interesting is that this plateau clearly happens between the 2 regimes: for large accumulated slip, the breakdown work follows the drained TP slip-on-a-plane solution, while at low accumulated slip, it follows the undrained/adiabatic behaviour.

Thank you for the suggestion, the caption of Figure 5 now specifies that the breakdown work is plotted at the same x/L_c position than in Fig. 3. The plateau between the two regimes caused by localisation is now compared two the standard TP model without localisation where the transition from the adiabatic undrained to the slip-on-a-plane regime is much smoother.

- Fig. 6 Prediction of the localization thickness (via Eq. 17). Taking a rupture speed $v_r/c_s = 0.6$, one can observe a spread between 0.07 and 0.3 for W_{loc}/h between curves. The simulation results are clearly not matching very well with the predicted curve. I would not call that well approximated - I am sorry - this is quite deceiving. Some additional simulations with different background shear stress / different values in a nucleation zone are also reported (with minimum context as to why they are needed)

This is actually the figure that relates the 1D prediction of [1] to the 2D elastodynamics. As discussed before, the applicability of the kinematic model with imposed slip rate to the description of dynamics rupture is not obvious. This prediction and related paragraph show how to apply the one dimensional theory to predict the thickness of the slipping layer of a dynamic event, including for ruptures that are far from steady-state due to the presence of stress heterogeneity. Despite all the simplifying assumptions, the prediction captures the trend observed in simulation over orders of magnitude change in thickness. Moreover, the simplicity of the prediction makes in an interesting tool for the interpretation of field measurements of the localisation thickness. The new version brought further context to this prediction and the results shown in Figure 6 at lines 295-308.

- One of the results put forward as important is that the fracture energy G_c estimated from the simulation is much smaller than the uniform shearing characteristic $\tau_c \delta_c$. However, within the assumption of a slip-on a plane model, the characteristic slip $\delta_c = \rho c h / (\Lambda f c)$ and shear stress $\tau_c = f_c \sigma'_0$ could be adapted by taking the actual width of the localized shear zone width (this is what is done in the slip-on a plane model where h has been taken from 'post-mortem' observations). One would get $\tau_c \delta_c = \rho c W_{loc} f_c \sigma'_0 / \Lambda$, from the results reported, it seems that such estimate would actually be much smaller than the reported G_c . It would have been nice to discuss this in details in the text.

Thank you for this relevant suggestion. Indeed, re-scaling the thickness of the actively slipping layer in the breakdown work predicted by adiabatic-undrained does not work to capture G_c measured in the simulation. The fact that it does not work underlines that the rupture energy is mainly associated to the rapid reduction in the size of the actively sheared layer—i.e. localisation—rather than by its consequence on the TP problem. We have included your suggestion in the discussion of the results at lines 276-284.

- I wonder if the authors have tried to compare their established profile of slip rate (Fig. 4) with results provided in Garagash (2012) (his fig 6 and the likes). Notably, I would imagine that studying steadily moving TP pulse accounting for strain localization would significantly help in better understanding end-member regimes, only “scratched on the surface” in the present manuscript.

We are only studying crack-like rupture in this paper. Studying slip pulse requires change in the nucleation process or implementing a dual-scale models based on the steady-state elastodynamics that is used in Garagash (2012). This could form the basis for follow up studies that we now highlight in the final sentences of the conclusion.

- I can't help but being left a bit disappointed by the lack of exploration of the dimensionless parameter space - could different regimes emerges with even less localization ? 2 simulations just do not cut it for my curiosity.

Our current nucleation procedure mostly allows two situations: either the perturbation is sufficient to initiate localisation-driven crack-like rupture or it is not and the shear zone returns to stable creeping conditions. An intermediate case with less localisation exists and correspond to the behaviour reported for the simulation with larger diffusivities. In this intermediate case, the first stage of the seismic rupture features mild localisation and the rupture dynamics is similar to the one driven by TP only. As the rupture accelerates, the slip rate increases and the localisation thickness drops, ultimately bringing the system into a localisation-driven dynamics. The selection of the regime is controlled by a nucleation length

reminiscent of what exists in slip-weakening and rate-and-state laws. As discussed above, studying the nucleation stage is a marvelous task that leaves the scope of this work and deserves a dedicated paper.

3 Section 5 discussion / final remarks

Shear strain localization (for an elasto-plastic rheology without contraction) is always associated with a loss of stress-bearing capacity (softening). Here the important point is that it is triggered by a multiphysics weakening mechanism: thermal-pressurization which has been well studied in the literature. The statement “shear localization controls the dynamics of EQ” is not strictly correct. Here it is the underlying weakening mechanism (Thermal pressurization) which is actually governing how the EQ evolves - eventually of course because the material “apparently” soften, shear strain do localize. It’s a sort of chicken and the egg problem - strain localization will not occur without strong weakening mechanism which are themselves further enhanced by localization. However here TP weakening is required for strain localization to kick in - so the latter is not the primary control.

Thank you for underlining this “chicken and the egg“ problem. We hope that the major revision of the document (detailed in the first part of the reply) have brought further support to the fact that localisation is the mechanism that leads to the emergence of crack-like brittle failure and drives them through a dynamics governed by a edge-localised dissipation G_c that is different from the dynamics of rupture driven by TP only.

The fact that multiphysics couplings brings complexity in the global energy balance & rupture dynamics while retaining features of LEFM near the tip has been found in numerous context - from hydraulic fracturing (Spence & Sharp 1985, Desroches et al. 1994) to underwater landslides (Viesca & Rice 2012) (not mentioning the EQ source literature). The supposedly strong statements of the paper are not really novel in my opinion. That said, the coupling of a local 1D thick thermo-poromechanics model with macroscopic elastodynamics of the rupture is a nice contribution and the conclusions of the paper confirm what can be easily anticipated from the existing body of knowledge.

We agree that our study enters into the broader class of problems where multiphysics at small scales controls the macroscale energy balance. This is already the case for TP without localisation (e.g. [3]). Our contribution is rather to highlight how strain localisation substantially affects the dynamics predicted from TP and that it could similarly affect the rupture dynamics in other geological context and rheology prone to develop strain localisation instability. We hope that the new simulations and the revision of the manuscript are better underlining how the coupled dynamics shows rather different results that what could be anticipated or obtained with standard model for thermal pressurization.

4 Minor Comments

The effect of thermal-pressurization on apparent frictional weakening was first recognized in rock slides - see (Habib 1967, 1975) - (Veveakis et al. 2007)

We thank the reviewer for pointing out these references. We actually realised that reference we originally gave for thermal pressurisation (Sibson 1975) was erroneous, and is now corrected to Sibson, 1973. To our knowledge, this was the first paper that actually computed the coupled temperature and pore pressure increase during slip. We could not access the paper by Habib 1967. In their 1975 paper, they only compute the temperature increase due to shear heating and simply conclude qualitatively that fluid vaporisation should lead to loss of strength. For this reason, and for the sake of brevity (our paper is not the place to retrace the intellectual history of the TP idea, however interesting this is), we will stick with the (now corrected) reference to Sibson's work.

- In Fig 3 panel b Wrsf in Eq.(10) ? Eq. 10 is not related with Wrsf . Is this a typo ? or I missed it completely.

Thank you, this has now been fixed.

- Having a more in-depth mechanistic presentation of the model & its scaling would really help to better understand the real novelty of the results. As it is written, it's extremely hard to decipher.

As previously mentioned, we have now presented the mechanistic aspects of the model in Section 3 of the revised manuscript, lines 136-150.

- quite a few typos in the reference list.

We went through the reference list several times to correct all possible typographical errors.

5 Reviewer #2

This is an extremely interesting paper presenting inspiring and original results on a key issue in earthquake mechanics, namely what controls earthquake dynamics during rupture propagation. I think that this work can potentially be of significance to the field and in related fields, such as fracture mechanics, after some (likely) substantial revisions. I am not completely convinced of the interpretation of the results to support conclusions and claims. It is not a matter of providing additional evidence, rather to balance the interpretations reconsidering some conclusions. The methodological contributions are rigorous and convincing; they are presented with excellent standards and a suitable approach. The methods are also presented with a sufficient detail ensuring reproducibility and a full understanding of the theoretical framework. I will present in the following my doubts on the provided interpretations and conclusions, hoping to be helpful and to clearly provide suggestions useful for the necessary revision of the manuscript. The authors propose the following conclusions:

- Earthquakes are likely to be systematically associated to extreme strain localization
- Strain localization controls dynamic weakening and creates an abrupt drop of shear stress promoting the propagation of classical dynamic ruptures (i.e., according to LEFM)
- The sudden localization of shear strain within a shear zone leads to the emergence of classical cracks driven by a constant fracture energy

I am personally convinced that strain localization plays a dominant role during faulting and earthquakes. It is strain localization that enables the creation of the fault core and the principal slipping zones (PSZs). This implies that strain localization is acting in different stages of the earthquake cycle, and it is not limited to the coseismic “additional” localization proposed in this study. Geological faults are characterized by fault cores with multiple PSZs. This implies that the strain localization process during individual earthquakes can be associated with different PSZs, likely suggesting heterogeneities of gouge materials which are not accounted by the simple theoretical definition of pressurized gouge used in this paper (i.e., constant shear stress in the fault-normal direction). This should be accounted by the authors by smoothing a bit the ambition to extrapolate their theoretical results to all earthquakes and all faults.

We thank the reviewer for the insightful review and enthusiasm about our work. We agree that strain localisation controls different aspects of faulting. To clarify the ambition of the work, we have now specified in the revised manuscript lines 69-71 that “In this paper, we focus on spontaneous strain localisation that occurs over short time scales—coincident to the few seconds that dynamic rupture propagation lasts—and leads to the formation of sub-millimetric principal slip surfaces within fault cores as sketched in Figure 1”.

I think that expressing shear stress evolution as a function of strain and strain rate represents a groundbreaking achievement in earthquake mechanics. Several papers in the literature attempted to tackle this challenge (see Beeler et al, JGR 1996), but a full representation of stress as a function of strain and strain rate is still missing. The authors try to address this issue but, according to my understanding, in this paper the shear stress is (solely) related to strain rate through sliding velocity. In other words, the dependence of shear stress on strain rate is not explicit. I would suggest clarifying the shear stress dependence on sliding velocity, since the latter controls strain localization, elastodynamics and thermal pressurization (through the heat source). It is unclear to me why strain localization dominates over others, as I will explain in the following.

Thank you, this is actually the outcome of the coupled model. The frictional response of the gouge in the dual-scale model proposed in the manuscript is only function of the strain rate. This is an important difference with the behaviour of TP models that prevent localisation and introduce, thereby, an explicit dependence on the slip rate. In the new version, this aspect has been clarified by including the simulation results from the TP model without localisation and by explicitly discussing the dependence to the strain rate in the text at lines 159-166.

My main concern about the interpretations applies to the description of the coupled dynamic system used to describe earthquake propagation. Immediately after nucleation along the creeping

fault, the rupture starts to propagate driven by elastodynamics (stress depends on sliding velocity as stated in equation 12), which is expected to evolve according to dynamic weakening. The adopted model includes thermal pressurization, which increases dynamic weakening, potentially leading to extreme dynamic weakening. Thermal pressurization requires a heat source and fluids; thus, it is activated by the initial dynamic weakening and slip acceleration. As the rupture propagates and slip velocity increases, strain is further localized within the fault zone because of the dependence of strain rate on slip velocity. Is this coseismic strain localization the effect or the cause of dynamic propagation? The authors interpret the result proposing strain localization as the cause, but I am not fully convinced by their reasoning. Because of its definition (strain is assumed to scale with slip and thickness), it is not surprising that increasing slip velocity increases strain rate, but the engine should be the elastodynamics and thermal pressurization that are the main drivers of dynamic weakening. In other words, I agree with the results shown in Figure 4, but they demonstrate the relation between slip velocity and strain rate, coherently with the adopted theoretical definition. I don't see these results as evidence of the dominant role of strain localization. What is the evidence corroborating that dynamic weakening is driven by strain localization and not by thermal pressurization?

This is an important point that has also been raised by the other reviewer. Please see the paragraph dedicated to the *chicken and the egg* question in the introduction of this reply for the list of changes made in the document. In short, the propagation of the simulated rupture is driven by strain localisation because

- Strain localisation introduces a plateau in the profile of the breakdown work that does not exist in the rupture dynamics of models that do neglect it.
- This plateau corresponds to the separation of scale between the dissipation at the tip (small slip) and at the tail (large slip) and leads to the emergence of crack-like dynamics in good agreement with LEFM theory.
- This plateau arises precisely at the same time—or slip—as the localisation instability.

The shear stress evolution with position (figures 2 and 4) and slip (figure 3) is quite interesting and, in my opinion, it deserves a careful interpretation. Looking at the slip weakening curves shown in figure 3 the shear stress initially raises to its peak value and the subsequent stress breakdown is divided in different stages with different slopes (stress gradient with slip). Can the authors identify and distinguish the contributions of thermal pressurization and strain localization during the dynamic breakdown stage? Can they prove that the abrupt weakening (shown in figures 3 and 4) is due to strain localization and not to thermal pressurization?

Yes, we can precisely identify in Figure 4 that the abrupt weakening at the origin of the plateau in the profile of breakdown work is directly correlated to strain localisation. Indeed both the abrupt drop of shear stress and the localisation of strain rate precisely arise between the positions B and C of Figure 4. The new version of the manuscript explicitly underlines this aspect between lines 254-265.

The association of slip velocity evolution with strain rate shown in Figure 4 does not corroborate, in my opinion, the dominant role of strain localization in controlling dynamic weakening. First, because this is the result of the analytical dependence of strain and slip rates. Second, because dynamic weakening inferred from stress evolution is very different from LEFM predictions. The authors just compare inferred shear stress and residual stress resulting from LEFM (or the initial raise before peak stress). Third, because peak slip velocity is reached at the end of the dynamic weakening stage. Dynamic weakening is associated with slip acceleration (points B and C in Figure 4), when strain is not yet fully localized. Strain localization is reached at the peak slip velocity (C in Figure 4), when dynamic breakdown is completed. Why is strain localization is considered the driver of dynamic weakening? We have now modified the axis of the main plot of Figure 4 to clarify this aspect. Regarding your three points:

1. As discussed above, the shear stress only relates to the strain rate in our model and this has been now mentioned in the text at lines 159-166.
2. The concentration of stress and velocity at the tip are actually quantitatively similar to the dynamics predicted by LEFM. This has been clarified by changing the vertical axis of the main plot to highlight that the profiles of stress and velocity can be fitted using a single LEFM asymptote and associated value of G_c . We have also included an additional figure (Fig. 7 in Appendix) that highlights that an equivalently good agreement are observed at other locations along the interface and lead to a similar value of G_c .
3. At point B, shear strain is homogeneous and at point C it is fully localised. The process of strain localisation arises then between B and C and directly corresponds to the dynamic weakening observed in the stress profile.

Interpreting figure 3, the authors state that predictions of shear stress at the tip $\tau_{sp}(d, Vtip)$ provide a good estimate of the residual shear stress reached after strain localization. I don't see such a corroborating fit and, more important, it is not clear to me why thermal pressurization should provide nearly constant or smoothly decreasing residual stress with slip (again, the authors should discuss why strain localization – interpreted as the driver of dynamic weakening, should overcome thermal pressurization) as shown in figures 3 and 4. I am also a bit surprised that, while slip velocity displays the expected behavior around the tip, the shear stress does not show any evident peak (as expected for a regularized singular field); rather it displays a very modest strength excess. This is not coherent with LEFM predictions, in my opinion. It might be also useful to compute the shear stress evolution inferred from thermal pressurization only (without strain localization) using it for comparison (also discussing the parameters' ranges in which thermal pressurization is weak or strong). The authors should better explain this behavior and discuss it to support their claims. Are we sure that this behavior is not due to the peculiar range of model parameter values used to simulate thermal pressurization?

Thank you for bringing the relevant suggestion of computing the shear stress evolution coming from TP only. This has led us to implement and run the second type of model that includes thermal pressurisation but prevents the possibility for strain localisation. We are convinced that the comparison between the rupture dynamics produced by these two models, shown in the new

Figures 7-8 and related discussion, brings important clarification and insights into the precise impact of strain localisation. We have also improved the plot of stress in Figure 4 to show how, despite the fact that the peak in shear stress is less sharp than the peak in velocity, both fields follow the same LEFM prediction with the same G_c . Finally, we fully agree that TP is not expected to produce a nearly constant residual stress with slip. It is actually strain localisation that produces it and this is a key difference with the model of TP that prevents it. We are explaining this aspect in the text (lines 254-265): If a clear plateau is observed in the breakdown work profile (Figure 5), the first weakening stage relevant for $\delta \ll \delta_c$ arises over a process region near the rupture tip much smaller than the second weakening stage, effective for $\delta \gg \delta_c$. In such case, the residual stress reached after the first weakening mechanism appears to be nearly constant at the scale of the process region (Figure 4), which explains the good agreement observed between the stress and velocity fields near the rupture tip and that predicted by dynamic fracture theory.

The interpretation of fracture energy and its scaling with slip during rupture propagation does not completely convince me. The authors claim that the good agreement with LEFM predictions shown in Figure 4 confirms the crack-like nature of the simulated rupture process near the tip of the slipping patch. They interpret this (debatable, see above) result to justify the computation of fracture energy from stress intensity factor (equation 15), governing the propagation of the dynamic rupture according to LEFM and small-scale yielding approximation (a first scale separation, see below). They therefore compute $G(d)$ (as defined by Abercrombie and Rice, 2005, equation 16) considering it the breakdown work in systems in which “frictional weakening” does not necessarily reach a well-defined residual value. However, looking at figures 3 and 4, the minimum shear stress to compute breakdown work following its definition (Tinti et al., 2005; Cocco et al., 2005) can be identified. The authors might also compute breakdown work. This is not a major issue and I understand that the authors implicitly distinguish fracture energy (tip dissipation) from frictional weakening. What is wondering me is which portion of shear stress evolution with slip during dynamic weakening corresponds to G_c from LEFM. The authors try to answer to this question interpreting figure 5, showing the scaling of breakdown energy (or breakdown work) with slip in different positions during the rupture propagation. Simulations with strong strain localization show a plateau corresponding to the G_c value. Can the authors interpret the range of slip values for which EBD is equal to G_c and the plateau appears? To which spatial positions these slip values correspond? Why there is only a given range of slip values for which EBD is equal to G_c ? What differs from smaller or larger slip values? Furthermore, looking at the shear stress evolution with slip, it is not surprising that breakdown work differs from fracture energy, at least because there is an initial hardening with non-negligible slip (i.e., there is mechanical work spent during strength excess).

We agree that there is different way of defining and computing the breakdown work. Here we choose this definition to match the one used in previous works that investigated dissipation for rupture driven by TP (e.g. [3]). The message of the paper is independent from this choice. The question of the separation of scales is very relevant and we have now clarified this in the text (both between lines 180-190 and 254-265). In a snapshot in time of the rupture, G_c is associated to dissipation in the spatial vicinity of the rupture tip, namely the process region that stretches between B and C in Figure 4. In a cross section in space, G_c is associated to the initial small slip distance of slip (few tenths of δ_c in Fig. 3c), whereas the remaining TP dissipation arises only once slip is larger than δ_c . This is the separation of scale that enables the LEFM crack-like dynamics—as formally

discussed in [7]— and that does not exist in absence of localisation for which the tip dissipation can no longer be isolated from the one at the tail.

The result shown in figure 5 is very interesting. To me, this confirms that breakdown work scales with slip during the dynamic propagation (as previously stated for pulse-like ruptures, Tinti et al., 2005; Rice et al., 2005; Cocco et al., 2006, among several others). If breakdown work is associated with frictional weakening, while constant fracture energy is associated with near-tip dissipation (I personally do not agree with this distinction, but this does not matter here), why they become equal only in a given slip range during rupture propagation? This result requires in my opinion a more comprehensive discussion. I am not asking to discuss the scaling of global estimates of fracture energy G' or breakdown work with total slip (i.e., a single value for each earthquake) as discussed by Cocco et al. (2023), because here we are focusing on the fracture energy and breakdown work values during dynamic propagation in different fault positions. Figure 5 clearly shows that a plateau appears when strain rate is localized, which means high slip rates. Can the authors discuss the dynamic propagation of a weakly localized rupture? Which slip velocity evolution is inferred in this case?

This relates to our reply to your previous comment and has now been addressed by including the simulation without localisation to emphasize the difference between the two rupture dynamics. Like the reviewer, we also do not agree with the systematic distinction between near-tip fracture energy and long-tail frictional dissipation. For some rupture, the near-tip dissipation cannot be isolated from the tail one and lead to rupture dynamics different than the one predicted by LEFM, like the dynamics reported in this manuscript for TP without localisation. Conversely, shear localisation creates a clear separation of scale and the rupture dynamics is similar to the one predicted by LEFM. As only the edge-localised energy controls the propagation in such case, this creates an important embrittlement of the fault.

The discussion of scale separation also requires some clarifications. The authors state that strain localization provides a clear separation between cohesive zone and the interior of the slipping patch, hence justifying the small-scale yielding hypothesis. This needs in my opinion clarifications. For classic LEFM predictions the scale separation concerns the zone of inelastic yielding near the crack tip and the large-scale elastic response of the bulk. Here, there is a scale separation between processes affecting strain localization within the thickness of shearing layer and along the crack length (transverse and longitudinal directions, as explained in the Appendix). Can the authors better explain why these different scales should be all coherent with the small-scale yielding hypothesis?

Thank you for pointing at these two separation of scales. We have followed your suggestion and clarified it in the text at lines 254-265.

The scale separation for LEFM is associated with a nearly constant residual stress independent of slip. Why strain localization should imply a nearly constant residual stress counterbalancing thermal pressurization, which on the contrary would imply a continuously decreasing shear stress with slip? The authors mention the cohesive zone only at the very end of the paper. According to elastodynamics, the size of the cohesive zone depends on slip velocity evolution. Once again, I remain with the doubt of which is the cause, and which is the effect. Everything turns around

the evolution of slip velocity, which occurs during dynamic weakening. It would be very useful to interpret the breakdown stage by analyzing the stress evolution, and not just the slip velocity profile. Strain localisation does not imply a constant residual shear stress. It just introduces a separation of scale such that, at the scale of the near-tip region, the residual stress looks constant (as visible in Fig.4) and can be described using LEFM. As discussed previously, this aspect has been clarified by the direct comparison that is now made with the rupture dynamics in absence of localisation.

In conclusion, I am convinced that the paper addresses a relevant issue in earthquake mechanics. The results might have a relevant impact for future investigations if interpretations are better explained and corroborated. Reading the paper, I was left with the impression that the reader is led to follow the interpretation preferred by the authors without having a complete understanding of the implications of this study. Perhaps this is just the narrative approach taken, but I remain of the opinion that the results should be discussed in a broader and more comprehensive way. For these reasons I recommend major revisions to strengthen the interpretation of the results justifying conclusions and claims.

Thank you for your positive comment. We hope that the revision has strengthened the interpretation of the results. The revised manuscript includes simulation using the benchmark TP model that prevents the possibility of localisation. The direct comparison with the rupture dynamics produced by our dual-scale model provides a broader perspective to the reader by discussing seismic rupture dynamics beyond the classical framework of LEFM.

References

- [1] J. D. Platt, J. W. Rudnicki, and J. R. Rice, “Stability and localization of rapid shear in fluid-saturated fault gouge, 2. Localized zone width and strength evolution,” *J. Geophys. Res.*, vol. 119, pp. 4334–4359, 2014.
- [2] J. R. Rice, J. W. Rudnicki, and J. D. Platt, “Stability and localization of rapid shear in fluid-saturated fault gouge, 1. Linearized stability analysis,” *J. Geophys. Res.*, vol. 119, pp. 4311–4333, 2014.
- [3] R. C. Viesca and D. I. Garagash, “Ubiquitous weakening of faults due to thermal pressurization,” *Nat. Geosci.*, 2015.
- [4] L. B. Freund, *Dynamic Fracture Mechanics*. Cambridge, UK: Cambridge University Press, 1990.
- [5] Y. L. Bai, “Thermo-plastic instability in simple shear,” *J. Mech. Phys. Solids*, vol. 30, no. 4, pp. 195–207, 1982.
- [6] S. Braeck and Y. Y. Podladchikov, “Spontaneous thermal runaway as an ultimate failure mechanism of materials,” *Phys. Res. Lett.*, vol. 98, no. 9, p. 095504, 2007.

- [7] E. Brener and E. Bouchbinder, “Theory of unconventional singularities of frictional cracks,” *J. Mech. Phys. Solids*, vol. 153, 2021.

Dear reviewers,

We first want to thank you for your valuable and enthusiastic feedback on our paper, which helped us improving it. Enclosed, please find the revised version of our manuscript entitled "Shear localisation controls the dynamics of earthquakes". We have addressed all your remaining comments and you can find, hereafter, a detailed reply where we have copied in blue font the original comments and appended our replies afterwards in black font. To easily identify the changes made to the document, a *latexdiff* pdf is submitted together with this document.

Yours Faithfully,

Fabian Barras, Nicolas Brantut

Reviewer #1:

I would like to apologize to the authors (as well as to the Editor & staff) for my late review. I had a chaotic agenda over the past 2 months which prevented me to dedicate sufficient time to carefully read the modified version of the manuscript and explanations provided by the authors to my original comments. I have now managed to do so today. I would like to thank the authors for having significantly re-shuffled the organization of the paper and for having performed a new set of simulations (without localization). I believe this clarify the effect of localization, and the analysis of the results (calculation of G_c et.) are now better described (as well as the results). I found the new version much clearer (may be bias also by being now more acquainted with the study). Notably, the authors have convincingly demonstrated the fact that shear-localization exhibit a clear isolation of an edge-localized fracture energy. I appreciated the re-wording of the different part of the papers (from model presentation to results & discussion) - This has greatly improved the importance of this study within the current body of knowledge. Nevertheless, I would be more at ease if the title would be e.g. "Importance of shear localization in the dynamics of EQs". Multiple physical phenomena are at stakes - there is not a single 'control'. But that's minor. I agree with the answers of my different initial comments, so I believe there is no need to go through all of them again. I have only very limited additional remarks.

- On the same lines that my suggestion for the modification of the title, I would rephrase the following phrase in the conclusion : "Importantly, the multi-scale model implemented in this work allows us to demonstrate quantitatively how localisation is the mechanism that drives the propagation of the earthquake rupture." to: "Importantly, the multi-scale model implemented in this work allows us to demonstrate quantitatively the importance of shear localisation on the propagation of earthquake rupture."

Following your suggestion, we have changed this sentence into "Importantly, the multi-scale model implemented in this work allows us to demonstrate quantitatively how localisation exerts a pivotal control on the propagation of the earthquake rupture" (lines 358-359). We chose not to change the title of the paper, considering that it does not imply a single control of localisation on the dynamics of earthquakes and that the multiple phenomena at stakes are further detailed in the abstract.

- New Figure 5. I believe there is an error in the labelling of the curves within the figure. Blue is with shear localization, Brown without ! not the opposite as written. (Also could the points of the curves be slightly made smaller ?- this would make the results more readable.)

Thank you for pointing out this mistake, which has been fixed in the new version. The marker size of the curves has also been reduced.

I would like to restate that the coupling of a local 1D thick thermo-poromechanics model with macroscopic elastodynamics of the rupture is a nice and novel contribution, whose impacts are clearly discussed and presented in this revised version. Provided that there are no changes to the text - beside possibly what I suggest above - I believe the paper can be published as is. Having the answers of the authors to the reviewers comments available to the interested readers would be a nice complement as well.

Reviewer #2:

The authors did an excellent job in revising the manuscript, addressing my previous comments and indications as well as responding to those of the other reviewer. I am convinced that this is an extremely interesting paper presenting inspiring and original results on key issues in earthquake mechanics, still debated in the community. The authors revised the manuscript in an efficient way, making the paper more accessible to readership, emphasizing the original results of their study making them more accessible. I believe that the paper is ready to be accepted for publication, but I would suggest a further check of the text and some minor improvement of the interpretations.

Here some indications that I hope will be useful to further improve the manuscript.

1. Comparison with classic dynamic crack propagation. The comparison with classic solutions from dynamic fracture mechanics is central to the paper. The authors use different terms to mention solutions from dynamic fracture mechanics, which might be misleading. They use "classical cracks driven by constant fracture energy" in the abstract (which is appropriate). Therefore, you mention LEFM and singular solutions at the tip of a dynamic fracture (rows 260-263 of the manuscript with track changes). Therefore, you mention "dynamic fracture solution (row 268) and you refer to the process zone (row 272) process region (row 289).

You refer to the crack-like nature of rupture dynamics (row 304). You cite the cohesive zone (row 341 and 444). You corrected the reference to LEFM in favor of dynamic fracture theory (row 365) and crack like rupture (row 427), but you maintain the predictions of LEFM in row 393. I would suggest referring to dynamic fracture theory (not LEFM, if not necessary) and cohesive zone. This for several reasons. First, your simulations do not include any singular field at the crack tip; therefore, a classic dynamic fracture theory with a cohesive zone (Palmer and Rice, 1973) well describe the comparison. Second, because you have a peak shear stress and a breakdown stress drop, and slip velocity is finite (not singular). In any case, I would recommend homogenizing the terminology.

Thank you for this suggestion. In the new version, we have reduced and homogenized the terminology related to dynamic fracture. In the text, "dynamic fracture" has been used instead of "LEFM" and the key concepts are now carefully introduced and defined (e.g. *fracture energy* line 62, *breakdown work* line 147, *process zone* line 150.)

2. The boundary integral formulation (equation 14). I am not sure it is appropriate to call the contribution of dynamic interactions ($\phi(x, t)$ in equation 14) "non-local dynamic contribution", because this open the question of the cohesive zone size and the slip velocity decrease to the residual value. I understand that you want to distinguish local dissipation, but you cannot exclude that your dual-scale model implies a cohesive zone size which makes relevant (i.e., non-negligible) the contribution from dynamic interactions in a region around the rupture front involving local and/or nearly local slipping points.

We have now clarified in the text (between lines 174-177) how this term differs from the other dynamic contribution of the equation. In the new version, this sentence reads "The integral on the right-hand side describes the instantaneous local stress change due to variations of the strain rate profile within the shear zone, whereas ϕ accounts for the dynamic interactions between different regions along the fault."

3. Separation of scales. In rows 211-214 you support the separation of scales making a reference to the parameterization of the problem (diffusive processes at the scale of the strain rate localization are smaller than the few meters-long distance between grid points). Is the scale separation depending on grid size (problem parameterization)? I don't think this is the case, but this sentence is misleading. I would suggest revising it.

We agree that this sentence is misleading. The scale separation does not depend on the grid size, it is rather the opposite: the relevant characteristic length scales set the grid size. This sentence has been changed into "The separation of scales is justified because the length scale of the diffusive processes over the duration of the seismic rupture is much smaller than the characteristic length scale over which stress and slip rate changes along the fault plane, typically of the order of a few metres." (line 187-190)

4. Interpretations of dynamic rupture solutions. There are still some interpretations that require, in my opinion, some check and clarification. I mention here some issues that I noticed but I would suggest a careful check by the authors.

- In figure 3 the residual stress does not seem to be constant and independent of slip. However, commenting Figure 4, you refer to a constant residual stress (rows 292-294).

We have removed this sentence referring to a constant residual stress.

- Your fracture energy estimates G_c are associated to a portion of the stress breakdown stage (and stress drop). Can you show this portion on the slip weakening curves (shear stress versus slip)? The dynamic weakening of shear stress shows a change of slope associated with the peak slip velocity, which should correspond to the transition between the edge-localized dissipation (according to your interpretation) and dissipation driven by thermal pressurization. Most of the breakdown stress drop occurs over the δ_c , which is associated with the strength reduction. I understand that the strain localization causes a dramatic reduction of the stress-bearing capacity, which promotes the subsequent strength reduction driven by thermal pressurization, but I would recommend to better explain this important aspect. I find this discussion still unclear in the manuscript.

Following your suggestion, we have highlighted the portion of the stress weakening curve in Figure 3 that corresponds to G_c . We are also showing and discussing the region of these plots where strain respectively localises and delocalises across the shear zone in order to better highlight how rapid slip acceleration, the associated shear stress weakening and G_c are all correlated to the sequence of strain localisation.

- Why the slip acceleration to peak slip velocity involves the same amount of slip and a nearly similar shear stress evolution around its peak? Your G_c are estimated through the comparison with elastic fracture mechanics, resulting from the comparison with simulations and estimates of elastic stress intensity factor (equation 17). Why this should work for a gouge material which is not elastic? Because of high strain rates caused by localization. This is a minor aspect for the revision, but it is not clear to me.

The fact that peak slip velocity and stress involve a similar amount of slip is self-selected by the coupling between the gouge rheology and the elastodynamics. It suggests that a characteristic slip distance should be accumulated to achieve weakening, in agreement with the results of 1D simulations from Platt, Rudnicki and Rice, JGR, 2014. This aspect is now further discussed between lines (210-215).

Regarding your second question, the applicability of linear elastic fracture mechanics is related

to the small-scale yielding conditions, which equivalently applies to the inelastic processes at the tip of fracture in brittle materials and the inelastic deformation of the gouge at the tip of seismic rupture. We have added the following lines to guide the reader through this concept: (lines 149-152) "Firstly, predictions of rupture dynamics often rely on the dynamic fracture theory and on the possibility to isolate a very small region around the rupture tip (the *process zone*) and associated near-tip energy dissipated in breaking the fault zone strength (the *fracture energy*).", (lines 259-262) "(...)in analogy to the small-scale yielding condition in brittle materials. This condition requires that the size of the process zone should be much smaller than other representative length scales of the elastic system for the propagation dynamics to be governed by the fracture energy G_c ." and (lines 268-279) "Such plateau in the profile of $E_{BD}(\delta)$ indicates that the rupture dynamics differs only slightly from the predictions of dynamic fracture theory and that the rupture propagation is well described by the fracture energy G_c , which explains the good agreement observed between the stress and velocity fields near the rupture tip and that predicted by dynamic fracture theory."

Minor corrections

- The legend of Figure 5 is wrong: the blue lines refer to strong strain localization
- The authors refer to equation (4.1) (rows: 331, 335, 402, at least), which is not included in the text. I guess this is equation (8) or Section 4.1? I am not sure. Please check.
- Substitute rupture with fracture at row 391. I guess you refer to fracture energy.

Thank you for pointing out these corrections, which have been implemented in the new version of the manuscript.

I hope these comments will be useful. I encourage you to make a further check and spend some more efforts to make your paper easily and fully accessible to readers. It is an extremely inspiring and timely contribution.

Yours Sincerely Massimo Cocco

Review of “shear localisation controls the dynamics of earthquakes“

December 4, 2023

The authors present a model of a planar fault under a mode III configuration (2D model) which account for the following physics:

- Elastodynamics of the solid material (which is thus assumed linear elastic) surrounding the fault,
- The “core” of the fault zone (the gouge) is modeled as a 1D fluid-saturated thermo-poro plastic material. This 1D model is in all points similar to the one of Rice et al. (2014), Platt et al. (2014). I will refer to it as the 1D-fault plane perpendicular model - or fault 1D gouge model hereafter. The gouge has a finite thickness and shear localization delocalization occurs within this initial gouge layer.
- Separability of scales is assumed between the “small” scale 1D-fault plane perpendicular model of the fault gouge and the macroscopic zero-thickness displacement discontinuity model of the rupture. The gouge model is coupled to the macroscopic model via the condition of homogeneous shear stress in the gouge and the definition of the macroscopic slip velocity as the integrated small-scale shear strain rate across the gouge.
- The fault gouge is assumed to exhibit a rate-hardening frictional rheology at critical state (i.e. without any inelastic porosity changes in response to strain). Therefore switching off the stabilization effect of dilation in the model of Rice et al. (2014), Platt et al. (2014). In this contribution, localization due to thermal-pressurization is therefore in sole competition with the rate-hardening frictional rheology.
- Homogeneous properties are assumed throughout.

The novelty of the paper is the coupling between a macroscopic equilibrium model (usual slip on a plane elastodynamics model of a fault) with the 1D finite thickness gouge model of Rice et al. (2014), Platt et al. (2014). The latter was solved under kinematically controlled conditions (imposing the strain rate) by Platt et al. (2014) and can thus be considered as a “local” although complex (time dependent etc.) constitutive behavior for the macroscopic problem. It is important to note that the coupling with macroscopic elastodynamics allow to solve for spontaneous rupture propagation. As the 1D gouge model strain localizes under some conditions usually met when Thermal Pressurization (TP) is active, it is not a surprise that one recovers macroscopic crack-like behaviors which are impacted by localization-delocalization due to the finite gouge thickness.

Some of the conclusions of the paper therefore already lays in the solution of the 1D finite thickness gouge model under kinematic conditions (Platt et al. 2014). In particular, the condition for a localization instability derived here is merely a re-expression of Rice et al. (2014) . The authors mention general “diffusive-like” effects (coupled to frictional rheologies) in section 2, but after focus solely on the classical thermal-pressurization (TP) phenomena: I would suggest to just focus on TP and state that the stability analysis is the one of Rice et al. (2014) (because this is what it is).

However, the coupled dynamics simulations presented in this paper are interesting in several aspects:

1) they provide a quantification of the effect of strain localization on the so-called ‘break-down’ work evolution with accumulated slip for such planar ruptures. (Although one should recognize that such evolution is mostly governed by the physics of TP).

2) they highlight a relation between rupture speed and the minimum thickness of the localized strain rates in the gouge (even though the derived approximation eq.(17) is not particularly convincing when compared to simulations).

1 Section by section comments & remarks

Section 2 - shear localization and faulting & section 3 - Model

As already mentioned, I am not a fan of presenting a “general” model coupling diffusive process with shear strain/stress when in fact the whole remaining paper focus solely on thermal pressurization (TP) and rate hardening friction. The ‘general’ linear stability analysis (detailed in appendix A) is nothing more than a mere recasting of previous works (Bai 1982, Rice et al. 2014). This must be clearly stated.

It would be best to have more discussion / precision on some of the assumptions of the model. Notably, the exact boundary conditions for T & p (I had to go in the numerical description in the appendix to fully understand them). It’s notably important as - from what I understand - the stability analysis assume a no-flux boundary conditions at the gouge boundary (so-called undrained /adiabatic limit), which is different than the numerical simulations which model diffusion in the surrounding medium perpendicular to the rupture. Basically, not all reader will be accustomed to the undrained/adiabatic and drained limits - which are key to understand the underlying effect of thermal-pressurization on fault slip.

An even more important point worth discussing in more details is when does the “separability” of scales actually “holds” (with respect to accumulated slip versus gouge thickness etc.). A priori, the coupling of a macroscopic slip along a planar zero-thickness macroscopic interface with the local 1D gouge model necessarily “forces” localization to happen and as result a crack-like behavior necessarily emerge. This must be recognized when analyzing the “generality” of the results.

Section 4 - results

As already mentioned, some features of the results are directly “built-in” the response of the 1D gouge model. I believe the results presented therefore must be compared with the ones of Platt et al. (2014) in much more details (Platt et al. (2014) provides the solution under kinematic control). As such, one can also obtain an evolution of the breakdown work as function of accumulated slip from the results of Platt et al. (2014). These results of Platt et al. (2014) can be post-process with the slip-rate history obtained here numerically to obtain another estimation of the break down work & this must be plotted against the results obtained here - e.g in Fig. 5. One would expect close agreement in most cases.

From the main text, it is difficult to grasp easily what is varied between simulations. It took me a while to understand (back and forth between main text, appendices and some thinking).The dimensional analysis in the appendix C indicates that the problem depends on 4 dimensionless quantities, and that they are taken representative of a fault zone at 7km depth for the “strong localization” simulation in the main text. Another simulation (weak localization) is performed by increasing the dimensionless hydraulic diffusivity 4 times higher (which is known to “tone” down the effect of TP). So 2 points in the dimensionless parameter space is probed. This is fine as the results are sufficiently rich, but the way the results are plotted & analyzed render their understanding quite difficult (at least to me).

Here is what I understood, and few remarks.

- Fig 2 highlights well the general evolution of the rupture as one would guess from the usual evolution of dynamic rupture and the results of Platt et al. (2014). Fig. 3 displays the quantities (macroscopic V and τ , micro-scale W_{loc}) as function of accumulated slip (macroscale quantities) at different location x/L_c along the 1D space. Lower x/L_c are positioned closer to the nucleation zone and therefore experience a rupture that has not completely ‘developed’/’localized’. The rupture is accelerating while developing in all the presented results.
- Fig. 4 - I understand very well how the stress intensity, residual strength and tip position are fitted from the profile of the slip rate in the moving tip coordinate system (and then G_c estimated). Nevertheless I was a bit puzzled as to i) when the profile is taken during the simulation (toward the end I assume?) ii) if the fit was performed on more than one profile (i.e at more than one time during the simulation), if the resulting value of G_c was varying ? I think that the profile is taken toward the end of the simulation, where the rupture has well developed toward a ‘crack’ like behavior away from the early-stage nucleation effect - but I could not be 100% sure from the manuscript. I imagine a sort of “goodness” of fit criteria for the LEFM model to apply can be reported as function of time to define when a well defined G_c emerges.
- Fig. 5 the breakdown work is reported for the same x/L_c position than Fig. 3 (this could be label in the Fig to help the readers). So for the strong localization, dark blue means position further away from the nucleation patch (and less “perturbed” by the initial phase of the rupture). A clear plateau at

constant G_c emerges. What is interesting is that this plateau clearly happens between the 2 regimes: for large accumulated slip, the breakdown work follows the drained TP slip-on-a-plane solution, while at low accumulated slip, it follows the undrained/adiabatic behaviour.

- Fig. 6 Prediction of the localization thickness (via Eq. 17). Taking a rupture speed $v_r/c_s = 0.6$, one can observe a spread between 0.07 and 0.3 for W_{loc}/h between curves. The simulation results are clearly not matching very well with the predicted curve. I would not call that well approximated - I am sorry - this is quite deceiving. Some additional simulations with different background shear stress / different values in a nucleation zone are also reported (with minimum context as to why they are needed).
- One of the results put forward as important is that the fracture energy G_c estimated from the simulation is much smaller than the uniform shearing characteristic $\tau_c \delta_c$. However, within the assumption of a slip-on a plane model, the characteristic slip $\delta_c = \rho c h / (\Lambda f_c)$ and shear stress $\tau_c = f_c \sigma'_o$ could be adapted by taking the *actual width* of the localized shear zone width (this is what is done in the slip-on a plane model where h has been taken from 'post-mortem' observations). One would get $\tau_c \delta_c = \rho c W_{loc} \sigma'_o / \Lambda$, from the results reported, it seems that such estimate would actually be much smaller than the reported G_c . It would have been nice to discuss this in details in the text.
- I wonder if the authors have tried to compare their established profile of slip rate (Fig. 4) with results provided in Garagash (2012) (his fig 6 and the likes). Notably, I would imagine that studying steadily moving TP pulse accounting for strain localization would significantly help in better understanding end-member regimes, only “scratched on the surface” in the present manuscript.
- I can't help but being left a bit disappointed by the lack of exploration of the dimensionless parameter space - could different regimes emerges with even less localization ? 2 simulations just do not cut it for my curiosity.

Section 5 discussion / final remarks

Shear strain localization (for an elasto-plastic rheology without contraction) is always associated with a loss of stress-bearing capacity (softening). Here the important point is that it is triggered by a multiphysics weakening mechanism: thermal-pressurization which has been well studied in the literature. The statement “shear localization controls the dynamics of EQ” is not strictly correct. Here it is the underlying weakening mechanism (Thermal pressurization) which is actually governing how the EQ evolves - eventually of course because the material “apparently” soften, shear strain do localize. It's a sort of chicken and the egg problem - strain localization will not occur without strong weakening mechanism which are themselves further enhanced by localization. However here TP weakening is required for strain localization to kick in - so the latter is not the primary control.

The fact that multiphysics couplings brings complexity in the global energy balance & rupture dynamics while retaining features of LEFM near the tip has been found in numerous context - from hydraulic fracturing (Spence & Sharp 1985, Desroches et al. 1994) to underwater landslides (Viesca & Rice 2012) (not mentioning the EQ source literature). The supposedly strong statements of the paper are not really novel in my opinion. That said, the coupling of a local 1D thick thermo-poromechanics model with macroscopic elastodynamics of the rupture is a nice contribution and the conclusions of the paper confirm what can be easily anticipated from the existing body of knowledge.

2 Minor comments

- The effect of thermal-pressurization on apparent frictional weakening was first recognized in rock slides - see (Habib 1967, 1975) - (Veveakis et al. 2007)
- In Fig 3 panel b W_{rsf} in Eq.(10) ? Eq. 10 is not related with W_{rsf} . Is this a typo ? or I missed it completely.
- Having a more in-depth mechanistic presentation of the model & its scaling would really help to better understand the real novelty of the results. As it is written, it's extremely hard to decipher.
- quite a few typos in the reference list

References

- Bai, Y. (1982), ‘Thermo-plastic instability in simple shear’, *Journal of the Mechanics and Physics of Solids* **30**(4), 195–207.
- Desroches, J., Detournay, E., Lenoach, B., Papanastasiou, P., Pearson, J. R. A., Thiercelin, M. & Cheng, A. (1994), ‘The crack tip region in hydraulic fracturing’, *Proceedings of the Royal Society of London. Series A: Mathematical and Physical Sciences* **447**(1929), 39–48.
URL: <http://rspa.royalsocietypublishing.org/content/447/1929/39.short>
- Garagash, D. (2012), ‘Seismic and aseismic slip pulses driven by thermal pressurization of pore fluid’, *Journal of Geophysical Research: Solid Earth* **117**(B4).
- Habib, P. (1967), ‘Sur un mode de glissement des massifs rocheux’, *Comptes Rendus Hebdomadaires des Seances de l’Académie des Sciences Série A* **264**(3), 151.
- Habib, P. (1975), ‘Production of gaseous pore pressure during rock slides’, *Rock mechanics* **7**, 193–197.
- Platt, J. D., Rudnicki, J. W. & Rice, J. R. (2014), ‘Stability and localization of rapid shear in fluid-saturated fault gouge: 2. localized zone width and strength evolution’, *Journal of Geophysical Research: Solid Earth* **119**(5), 4334–4359.
- Rice, J. R., Rudnicki, J. W. & Platt, J. D. (2014), ‘Stability and localization of rapid shear in fluid-saturated fault gouge: 1. linearized stability analysis’, *Journal of Geophysical Research: Solid Earth* **119**(5), 4311–4333.
- Spence, D. & Sharp, P. (1985), ‘Self-similar solutions for elastohydrodynamic cavity flow’, *Proceedings of the Royal Society of London. A. Mathematical and Physical Sciences* **400**(1819), 289–313.
- Veveakis, E., Vardoulakis, I. & Di Toro, G. (2007), ‘Thermoporomechanics of creeping landslides: The 1963 vaiont slide, northern italy’, *Journal of Geophysical Research: Earth Surface* **112**(F3).
- Viesca, R. & Rice, J. (2012), ‘Nucleation of slip-weakening rupture instability in landslides by localized increase of pore pressure’, *Journal of Geophysical Research* **117**(B03104), 21.

Review of the revision of “shear localisation controls the dynamics of earthquakes“

August 24, 2024

I would like to apologize to the authors (as well as to the Editor & staff) for my late review. I had a chaotic agenda over the past 2 months which prevented me to dedicate sufficient time to carefully read the modified version of the manuscript and explanations provided by the authors to my original comments. I have now managed to do so today.

I would like to thank the authors for having significantly re-shuffled the organization of the paper and for having performed a new set of simulations (without localization). I believe this clarify the effect of localization , and the analysis of the results (calculation of G_c et.) are now better described (as well as the results). I found the new version much clearer (may be bias also by being now more acquainted with the study).

Notably, the authors have convincingly demonstrated the fact that shear-localization exhibit a clear isolation of an edge-localized fracture energy. I appreciated the re-wording of the different part of the papers (from model presentation to results & discussion) - This has greatly improved the importance of this study within the current body of knowledge. Nevertheless, I would be more at ease if the title would be e.g. “Importance of shear localization in the dynamics of EQs”. Multiple physical phenomena are at stakes - there is not a single 'control'. But that's minor.

I agree with the answers of my different initial comments, so I believe there is no need to go through all of them again. I have only very limited additional remarks.

- On the same lines that my suggestion for the modification of the title, I would rephrase the following phrase in the conclusion :”Importantly, the multi-scale model implemented in this work allows us to demonstrate quan-titatively how localisation is the mechanims that drives the propagation of the earthquake rupture.” to: ”Importantly, the multi-scale model implemented in this work allows us to demonstrate quantitatively the importance of shear localisation on the propagation of earthquake rupture.”
- **New Figure 5. I believe there is an error in the labelling of the curves within the figure. Blue is with shear localization, Brown without ! not the opposite as written.** (Also could the points of the curves be slightly made smaller ?- this would make the results more readable.)

I would like to restate that the coupling of a local 1D thick thermo-poromechanics model with macroscopic elastodynamics of the rupture is a nice and novel contribution, whose impacts are clearly discussed and presented in this revised version. Provided that there are no changes to the text - beside possibly what I suggest above - I believe the paper can be published as is. Having the answers of the authors to the reviewers comments available to the interested readers would be a nice complement as well.